# Anterior cingulate cortex provides the neural substrates for feedback-driven iteration of decision and value representation

Wenqi Chen[1,3], Jiejunyi Liang [2,3], Qiyun Wu[2] & Yunyun Han [1] ✉

Adjusting decision-making under uncertain and dynamic situations is the hallmark of intelligence. It requires a system capable of converting feedback information to renew the internal value. The anterior cingulate cortex (ACC) involves in error and reward events that prompt switching or maintenance of current decision strategies. However, it is unclear whether and how the changes of stimulus-action mapping during behavioral adaptation are encoded, nor how such computation drives decision adaptation. Here, we tracked ACC activity in male mice performing go/no-go auditory discrimination tasks with manipulated stimulus-reward contingencies. Individual ACC neurons integrate the outcome information to the value representation in the next-run trials. Dynamic recruitment of them determines the learning rate of error-guided value iteration and decision adaptation, forming a non-linear feedback-driven updating system to secure the appropriate decision switch. Optogenetically suppressing ACC significantly slowed down feedback-driven decision switching without interfering with the execution of the established strategy.

The ability to adapt one's behavioral strategy in complex environments is at the core of cognition. Doing it so efficiently requires continuously monitoring the outcomes of actions, evaluating the ongoing strategy, and, when appropriate, switching away from it[1]. For example, in typical goal-directed behaviors, such as the sensory discrimination go/no-go task, the value of the cue stimulus, rather than the properties of the cue itself, is extracted for decision-making (Fig. 1a). Meanwhile, the decision generates an internal prediction of the action, and externally, the execution of the decision results in an actual outcome[2]. The difference between those two is detected as prediction error ($\Delta r$) and used for updating the value representation of the stimulus in the next-run decision process, forming an internal feedback loop essential for decision adjustment. When the stimulus-reward contingency differs, the error experience ($\alpha\Delta r$) is cumulated and carefully analyzed to derive the new relationship of action and outcomes, and eventually, forms a new optimal decision strategy. Where and how is such an internal feedback loop

carried out? Studies in humans, primates, and rodents have shown that anterior cingulate cortex (ACC) neurons respond to reward expectation[3] and omission[4], suggesting that the ACC encodes prediction- and error-related information[5–9]. Patients with a damaged ACC exhibit impaired judgment, planning, and decision-making[10,11]. Optogenetic inhibition of the rodent ACC significantly decreases task state updating and prediction[12,13]. Therefore, information on cognition control required for flexible behavior is heavily processed in the ACC, but it is still not clear how the information is computed to derive a new decision strategy. Specifically, how is the change of stimulus-reward contingency encoded in ACC? How is such information computed and used to adjust decision strategy? And how are neural resources dynamically recruited according to cognition demands?

To address these questions, we used two-photon calcium imaging to track the activity of the ACC neurons while mice performing go/no-go auditory discrimination tasks with the progressive change of

[1]Department of Neurobiology, School of Basic Medicine, Tongji Medical College, Huazhong University of Science and Technology, Wuhan 430030, China. [2]State Key Laboratory of Intelligent Manufacturing Equipment and Technology, Huazhong University of Science and Technology, Wuhan 430074, China. [3]These authors contributed equally: Wenqi Chen, Jiejunyi Liang. ✉e-mail: yhan@hust.edu.cn

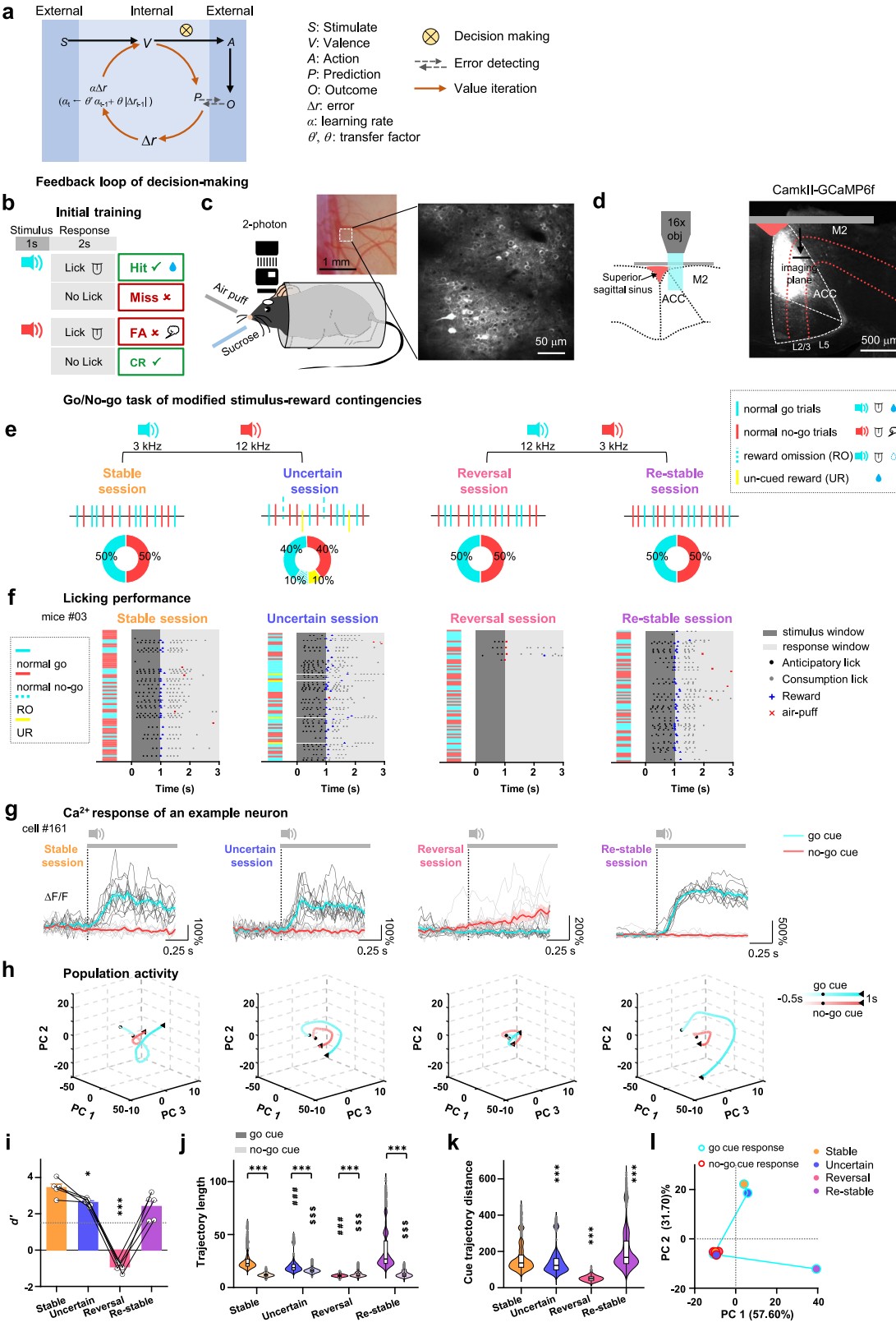

stimulus-reward contingencies. Individual ACC neurons were able to detect and convert error feedback of behavior outcome to the value representation of the task-relevant stimulus, and drove the shift decision strategy and ACC activity. Further suppressing ACC activity slowed down the feedback-induced decision switch without hurting the correct choice following the previously established decision strategy.

## Results

### Modulation of stimulus-reward contingencies drives the drift of value representation of task-relevant stimulus in ACC

We used a series of go/no-go auditory discrimination tasks to test how the animals responded to various unexpected changes in stimulus-reward contingencies. First, we trained head-fixed mice performing the go/no-go auditory discrimination task[14] (Fig. 1b). Each trial started

**Fig. 1 | Variation of stimulus-reward contingency drives the shift of decision and stimulus representation in ACC neurons. a** Hypothetic process of optimizing decision making during goal-directed behavior. **b** Go/no-go auditory discrimination task structure and outcomes during initial training. CR correct rejection, FA false alarm. **c** In vivo two-photon imaging of ACC neurons ($n = 6$ mice). **d** Scheme (left) of imaging L2/3 ACC neurons through a cranial window (grey horizontal bar) and a coronal section (right) of ACC neurons expressing GCaMP6f in the recorded mouse ($n = 6$ mice). M2: the secondary motor cortex. **e** Schemes of four go/no-go tasks with modified stimulus-reward contingencies. **f** Raster plots of the licking behavior of an example mouse in all task sessions. **g** Ca$^{2+}$ transients of an example ACC neuron in response to go and no-go stimuli in each session. **h** Population activity trajectories in the PCA space, explaining ≥66% of variance (493 cells from 5 mice). Black dots /triangles: stimulus onset/offset. **i** Performance discrimination ($d'$) across sessions. Dashed line: performance threshold ($d' = 1.5$). **j** Euclidean length of activity trajectories during the stimulus window in each task session. Symbol * indicates difference between go and no-go cue evoked responses within the same session (two-sided Wilcoxon rank-sum test). # or $ indicates go or no-go differences compared to the Stable session. **k** Euclidean distance between activity trajectories of go and no-go responses in each task session. **l** Relative distance of the population activity of go and no-go response of each task session. *$P < 0.05$, ***$P < 10^{-3}$, $^{###}P < 10^{-3}$, $^{$$$}P < 10^{-3}$. Data analyzed by (**i**, $n = 5$ mice) two-sided one-way RM ANOVA with post-hoc Dunnett's comparisons, or (**j**, **k**: $n = 5,000$ bootstrap iterations) two-sided Kruskal-Wallis test with post-hoc Bonferroni comparisons, comparing to the Stable session. Data are presented as (**g**, **i**) mean ± s.e.m. or (**j**, **k**) violin plots (the shape represents individual distribution; center line, median; box limits, upper and lower quartiles; whiskers, 1.5 × interquartile range). Statistical details are presented in Supplementary Table 1. Source data are provided as a Source data file.

with a pure tone stimulus of 3 kHz or 12 kHz lasting for 1 s, followed by a 2-s response window. Mice were rewarded with sucrose solution for licking the spout in response to the go cue (3 kHz), or punished with an air puff for licking in response to the no-go cue (12 kHz). Reward and punishment were only delivered upon the first lick in the response window. Mice typically learned the task rapidly within 3-4 days (Supplementary Fig. 1a), from licking indiscriminately upon both cues to almost exclusively licking upon the go cue (Supplementary Fig. 1b).

After the mice achieved steady high performance after initial training, the Ca$^{2+}$ dynamics of ACC excitatory neurons in layer 2/3 were recorded by GCaMP6f (Fig. 1c, d) through a series of sessions consisting of modified go/no-go paradigms with progressively shifted stimulus-reward contingencies (Fig. 1e). The Stable session consisted of 50/50% of randomized go and no-go trials. In the Uncertain session, two additional types of perturbing trials were introduced: reward omission (RO, 10% of total) trials, in which the reward was omitted for correct licking upon go cue (3 kHz) during the response window, and un-cued reward (UR, 10% of total) trials, in which sucrose reward was given without any sensory stimulus. Thus, the unpredictable reward omission and un-cued reward weakened the stimulus-reward association. In the Reversal session, stimulus-reward contingencies were reversed, that is, licking upon a 3 kHz tone would be associated with punishment instead of reward and vice versa. Such changes were totally unexpected at the beginning of the Uncertain and Reversal session, thus the discrimination performance significantly dropped (Fig. 1f, i). After several sessions of reverse learning, the animal again achieved high discrimination performance ($d' ≥ 1.5$) in the Re-stable session with the new stimulus-reward association (Supplementary Fig. 1a–c).

We tracked the activity of ACC neurons ($n = 493$ neurons from 5 mice, Supplementary Fig. 2a) through all four sessions and analyzed their response during the stimulus window, focusing on whether and how the differential representations of go and no-go cues were dynamically modulated when the stimulus-reward contingencies were manipulated. The reward-associated tone evoked a much stronger response than the no-go stimulus (Fig. 1g and Supplementary Fig. 2b, c). The response during the stimulus window was not likely caused by licking upon the go cue, since the response was much earlier than the licking onset in the stimulus window, and neither the onset nor amplitude of neural response correlated with licking (Supplementary Fig. 3). We analyzed the principal component of the population activities during stimulus window[15,16] (Fig. 1h). Trajectories of go cue-evoked population activity expanded over much broader PCA space than those of the no-go cue in all sessions except Reversal (Fig. 1j), indicating the ACC neurons preferably encoded more information about the reward-associate stimulus. The cumulative distance between trajectories of go and no-go response decreased significantly in the Uncertain and Reversal sessions (Fig. 1k), and re-emerged again in the Re-stable session, indicating that population discrimination of go and no-go cues strongly correlated with the stimulus-reward

association. Further analyzing the first two PCA components of ACC population activity showed that the no-go cue evoked response in the four stages clustered together in the third quadrant, clearly separating from the go cue evoked response except in the Reversal session when the same tone was previously associated no-go cue (Fig. 1l). The go cue evoked responses of the Stable, Uncertain, and Re-stable sessions were located in the first and fourth quadrants, suggesting that the PC1 might represent the value of the cue stimulus, whereas the PC2 might represent the feature of the reward-associated stimulus. The change of stimulus-reward contingency pushed the drift of population activity.

## The cumulation of unexpected outcomes is dynamically encoded in the ACC population

To gain insight into how the stimulus representation of ACC changes with behavior strategy upon experiencing modification of stimulus-reward association, we compared the ACC activity during the Stable and Uncertain sessions. When subjects encounter unexpected changes in their familiar environment, they usually follow previous routines until cumulating enough evidence to support a new strategy[17–19]. Compared to the Stable session, the same 3 kHz tone was still associated with a reward in the Uncertain session, but the probability of such stimulus-reward coincidence went down because of the unexpected outcomes in 20% of trials (RO & UR, Fig. 1e and Supplementary Fig. 4a). The mean responses of the ACC neurons to go and no-go cues remained largely unchanged, but the overall task performance slightly decreased (Fig. 2a). Notably, the performance at the early stage of the Uncertain session was quite steady and similar to the Stable session, but the miss trials started to increase in the late stage of the session (Fig. 2b, c, and Supplementary Fig. 4a). We divided the Uncertain session into three phases (T1-T3, corresponding to the first, middle, and last 33% trials, respectively), and compared them to the Stable session. The FA rate remained consistently low, whereas the Hit rate was significantly reduced in the T3 phase (Fig. 2c). The mean responses to the go cue only slightly decreased in the T3 phase (Fig. 2d). Interestingly, the go stimulus-evoked ACC population activity was significantly down-regulated when the outcome of the previous trial was unexpected (Fig. 2e and Supplementary Fig. 4d), but the behavior decision remained unaffected. It suggested that just one unexpected outcome was able to trigger an immediate change in the value representation but not enough to change the established decision-making rule. Decision strategy was more resilient to a single error incident, and required to collect several unexpected events to conclude the necessity to change. The experience of RO trials led to an adjustment in the decision strategy, but not a decrease in motivation to perform the task, as the lick latency, the number of anticipatory and the consumption lick rate were stable throughout T1-T3 phases in the Uncertain session and as high as in the Stable session (Supplementary Fig. 5). There must be a system that could detect unexpected outcomes, store and integrate the history of such experiences, and evaluate it to choose whether to change the decision-making strategy.

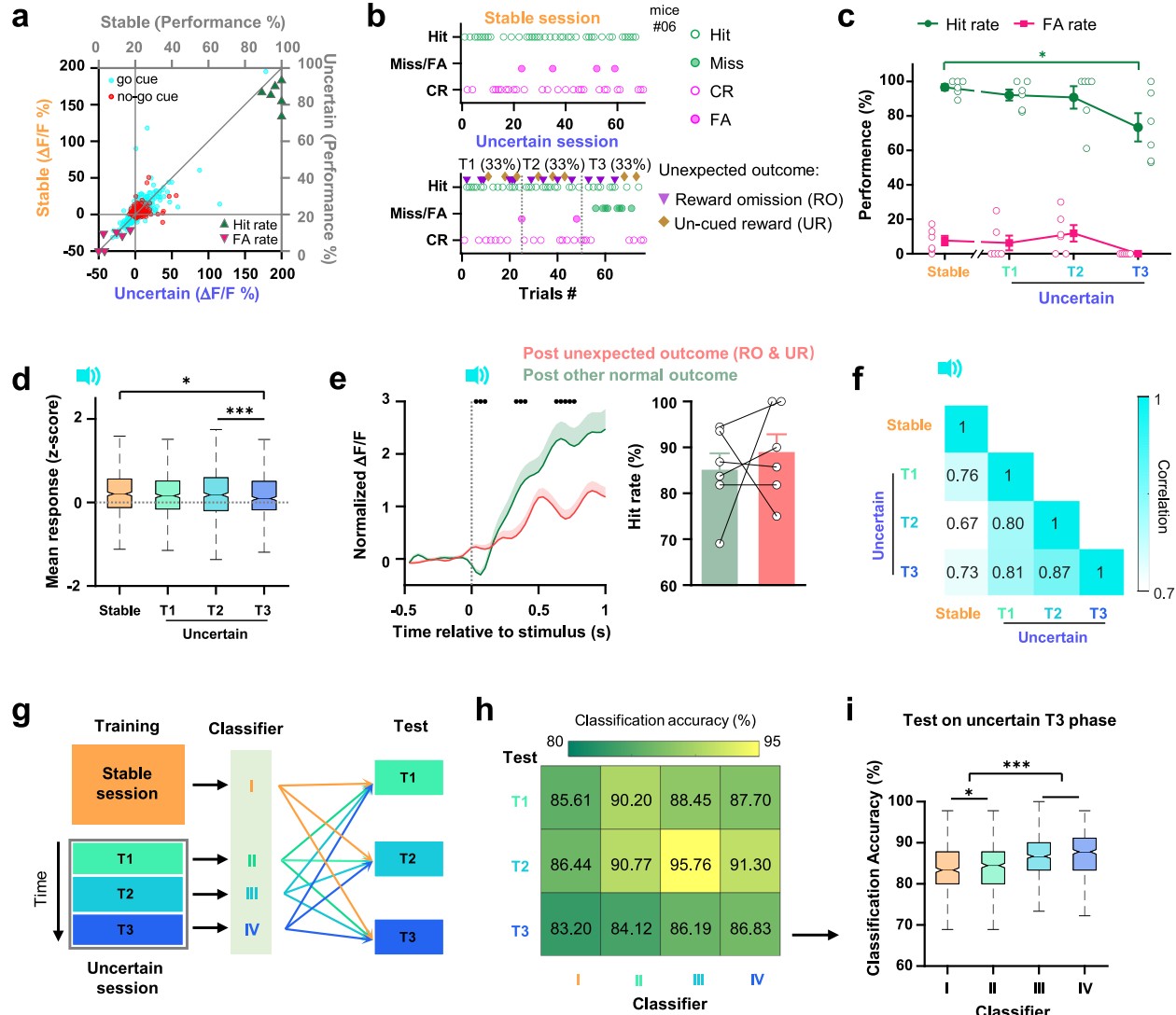

**Fig. 2 | The cumulation of unexpected outcomes drives changes of decision and population representation of stimulus. a** Scatter plot of the mean stimulus response (mean ΔF/F during stimulus window) of each neuron across the Stable and Uncertain sessions (n = 549 neurons from 6 mice). **b** Behavioral performance from an example animal during the Stable (top) and Uncertain (bottom) sessions. Inverted triangle and diamond markers indicate unexpected outcomes in the Uncertain session, divided into three phases (T1-T3) by vertical dashed lines. **c** Behavioral performance in the Stable session and T1-T3 phases of the Uncertain session. (n = 6 mice, two-sided one-way repeated measures ANOVA with post-hoc Dunnett's comparisons, comparing each phase to Stable session). **d** Mean population response to go stimulus in the Stable session and T1-T3 phases in the Uncertain session (n = 549 neurons from 6 mice, two-sided Friedman test with post-hoc Bonferroni comparisons). **e** Left, average population responses to go stimulus in trials following unexpected outcomes (red, including RO and UR trials) and

following other normal outcomes (green). Black dots indicate the time segments when the activity is different from each other (P < 0.05, two-sided Wilcoxon rank-sum test). Right, the Hit rate of trials following unexpected outcomes and other normal outcomes (n = 6 mice, two-sided paired t-test). **f** Correlation of go-cue evoked population activity between the Stable session and T1-T3 phases in the Uncertain session. **g** Scheme of training and test classifiers to decode the task stimulus with population activity. **h** Accuracy of classifying stimulus identities from population neuronal activity within and across different task phases. **i** Classification accuracy of neural activity in the T3 phase using classifiers trained from neural activity of different phases (n = 500 times repeat, two-sided Kruskal-Wallis test with post-hoc Bonferroni comparisons). *P < 0.05, ***P < 10⁻³. Data are presented as (**c, e**) mean ± s.e.m. or (**d, i**) box plots (center line, median; box limits, upper and lower quartiles; whiskers, 1.5 × interquartile range). Statistical details are presented in Supplementary Table 1. Source data are provided as a Source data file.

The correlation of go cue-evoked activity between the Stable session and the three phases of the Uncertain session decreased with time (Fig. 2f), indicating that a gradual shift of stimulus representation in ACC might be a result of a continuous updating loop driven by the outcome feedback over time. To test it, we trained four classifiers to decode stimulus identity from population activity of the trials in the Stable session (Classifier I) and the T1-T3 phases of the Uncertain session (Classifiers II-IV, respectively), and tested their prediction accuracy with the experiment data collected in each (T1-T3) phase (Fig. 2g). Stimulus identities were best decoded by the classifier generated from the same task phase and worst decoded by the classifiers

generated from the furthermost phase or the Stable session (Fig. 2h, i and Supplementary Fig. 4k). It demonstrated that the feedback of unexpected outcomes not only efficiently modulated the population representation in ACC, but their influence was also well accumulated, stored as the drift of value representation throughout the entire task, and in the end, led to the change of decision strategy.

## Feedback-driven iteration of value representation is carried out by individual ACC neurons

How were the unexpected outcomes converted to the signal to guide the shift of stimulus representation and decision-making strategy?

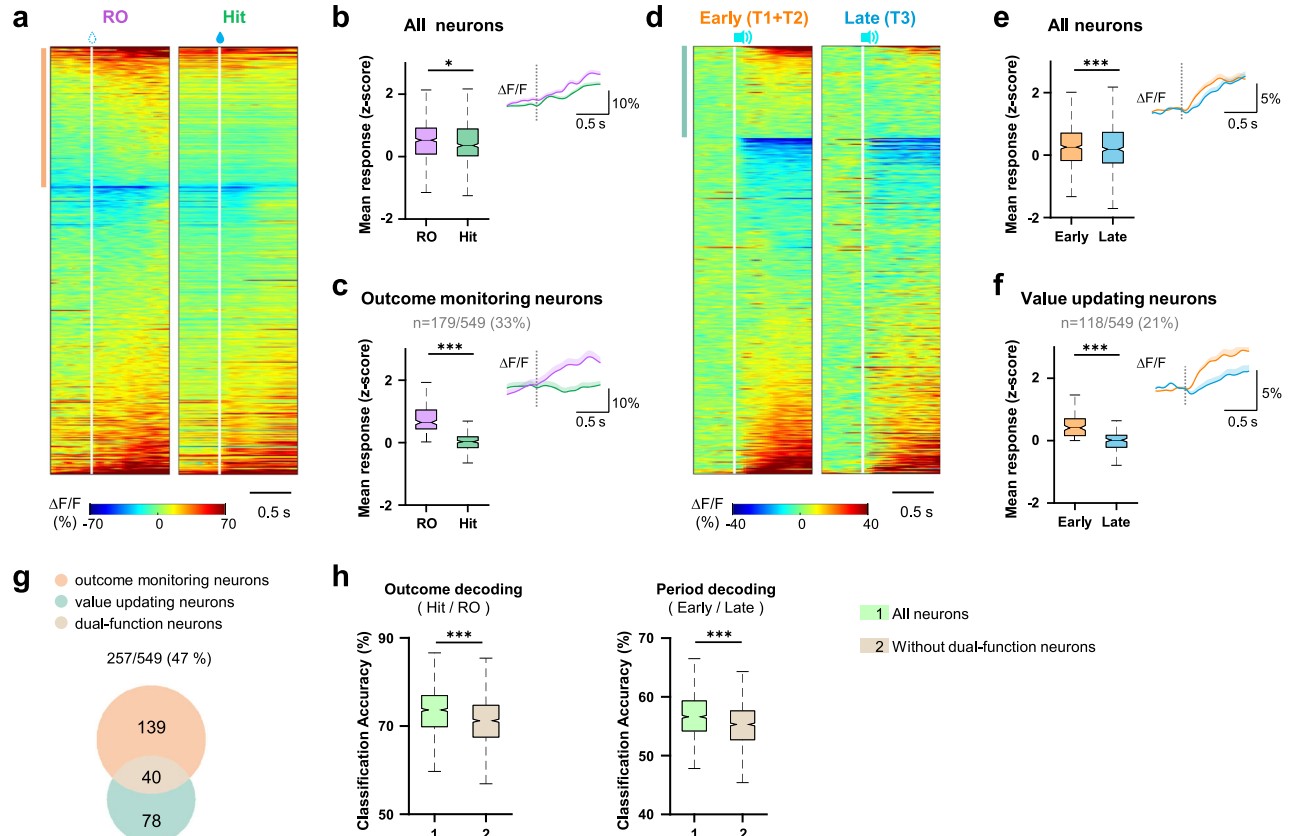

**Fig. 3 | Individual dual-function ACC neurons temporally integrate outcome information to the next-run value representation. a** Trial-averaged activity of all neurons in reward omission (RO, left) and Hit (right) trials. White vertical lines denote reward delivery time. ($n$ = 549 neurons from 6 mice). Each row represents one neuron. **b** Mean population responses of all neurons during 1-s window after reward omission or delivery during RO and Hit trials. Inset: Trial-averaged Ca$^{2+}$ response (mean ± s.e.m.). Dashed vertical lines denote reward delivery time. **c** Mean population responses of identified outcome monitoring neurons (marked by light orange vertical bars adjacent to heat maps in **a**, similar to **b**). **d** Trial-averaged go cue-evoked response of all neurons in the early (T1 + T2, left) and late (T3, right) periods. White vertical lines denote the stimulus onset. **e** Mean population responses of all neurons during 1-s window after cue onset in early and late periods. Inset: Trial-averaged Ca$^{2+}$ response (mean ± s.e.m.). Dashed vertical lines denote the stimulus onset. **f** Mean population responses of identified value updating neurons (marked by light blue bars adjacent to heat maps in **d**, similar to **e**). **g** Venn diagram of the number of neurons identified to perform outcome monitoring (light orange) and value updating (light blue) functions. **h** Accuracy of decoding outcome (left) and period (right) using a subpopulation of 200 neurons randomly selected from all ($n$ = 549) neurons or all but without dual-function ($n$ = 509) neurons. *$P$ < 0.05, ***$P$ < 10$^{-3}$. Data analyzed by (**b, e** $n$ = 549 neurons; **c** $n$ = 179 neurons; **f** $n$ = 118 neurons, from 6 mice) two-sided Wilcoxon signed-rank test, or (**h**: $n$ = 1,000 bootstrap iterations) two-sided Wilcoxon rank-sum test. In box plot: center line, median; box limits, upper and lower quartiles; whiskers, 1.5 × interquartile range. Statistical details are presented in Supplementary Table 1. Source data are provided as a Source data file.

ACC could be featured with a network with temporal integrating function, via which the unexpected outcomes could not only be detected but also cumulated to trigger the shift in the population representation and decision. Such systems required two components: (1) monitoring the outcome and detecting unexpected events; and (2) storing and integrating the history of unexpected outcomes to the representation of stimulus value. We examined ACC neurons with such functional signatures. The unexpected reward omission-evoked response (1 s after the first lick in the reward window) was stronger than the predicted reward-evoked response (Hit trials, Fig. 3a, b), indicating that the outcome was closely monitored and the unexpected incident was immediately detected in the ACC. We identified neurons with remarkably increased activity when experiencing reward omission as outcome monitoring neurons (33% of all recorded ACC neurons, Fig. 3c), which were also able to detect the unexpected uncued rewards (Supplementary Fig. 6).

To examine how the representation of stimulus evolved through the Uncertain session, we compared the early (high value, T1 and T2 phases) and late (low value, T3 phase) periods. The relative decrease of response to the go stimulus in the late period represented the gradual diminution of the positive value of the go cue, which could be an accumulative value re-assessment of go cue-related decisions (Fig. 3d, e). Hence, we defined those neurons with smaller go cue-evoked responses in the late period as value updating neurons (21% of all recorded ACC neurons, Fig. 3f). During the Uncertain session, outcome monitoring and value updating neurons together accounted for about half of all recorded ACC neurons. More interestingly, a small portion of them were dual-function neurons that carried out both outcome monitoring and value updating functions (Fig. 3g).

We trained a new set of classifiers to decode the outcome and task phase, and test the decoding accuracy of the whole population ($n$ = 549 neurons) and the subpopulation without the dual-function neurons ($n$ = 509 neurons). Although the dual-function neurons were only a small proportion here (40 out of 549 neurons), the decoding accuracy dropped significantly when they were removed from the population (Fig. 3h), indicating that they played an important role in computing outcomes and tracking experience history. The dual-function neuron might be a natural computation and storage element to integrate the outcome information to the value representation of the stimulus by regulating their single-neuron plasticity. Each dual-function neuron can carry out the iterations of value updating, forming the fundamental block of the system of evidence collecting and

evaluating. Together they could work as parallel computation units to accelerate the iterations of feedback driven value updating necessary for decision adjustment.

## Recruitment of dual-function neurons forms a non-linear value iteration system to control the learning rate

If the subject needed to accumulate several unexpected events to shift the value representation to the threshold to finally change the decision-making strategy, the switching could be vigorously accelerated in reverse learning (Figs. 1i and 4a). Consecutive punishments due to licking upon the previous go cue (3 kHz tone) at the beginning of the Reversal session produced a greater degree of surprise and a higher probability of unexpected outcomes (Fig. 4b, c). Compared to the Stable session, the previous go cue (3 kHz tone) evoked neuronal response and licking behavior was drastically suppressed (Fig. 4b), indicating a process of devaluation of the previous go stimulus and dissociation of the licking decision from it. As we predicted, the most drastic change in behavior decision happened at the very early phase when the animal was consecutively exposed to unexpected punishment (Fig. 4c and Supplementary Fig. 8a). Therefore, we divided the Reversal session into three phases, corresponding to the first 15%, middle 35% and last 50% of the total trials (T1-T3, respectively). The animals rapidly managed to down-regulate the neuronal response and to switch off the licking behavior just after a few air-puff punishments in the early phase (T1, Fig. 4c, d). Experiencing a single FA trial could immediately decrease the tone evoked response in the adjacent no-go trial, but not sufficiently to change the decision (Fig. 4e), indicating that, even in a scenario as drastic as the reverse learning, adjustment of decision-making strategy still robustly relied on the accumulation of experience rather than a single error event. The correlation of population response to the 3 kHz stimulus between the Stable and Reversal sessions was low, and further dropped down with time (Fig. 4f).

Are the outcome monitoring and value updating functions carried out in the same way in the rapid decision changing process? We compared the early (T1) and late (T2 and T3) periods of the Reversal session. In the early period, since animals still took the 3 kHz tone as the reward-associated cue, the air-puff punishment was most unexpected. The punishment evoked population response (1 s since air-puff delivery) was stronger in the early period (Fig. 4g, h), revealing that a more prominent error signal was generated when the outcome was most unexpected. Here we defined those neurons with remarkably stronger responses to punishment in the early period as outcome monitoring neurons (Fig. 4i). On the other hand, the no-go stimulus-evoked response was also stronger in the early period, because the previous reward-associated stimulus was devaluated by repeated punishment (Fig. 4j, k). Those neurons with lower response to the no-go stimulus in the late period were defined as value updating neurons (Fig. 4l). The majority of neurons retained their functionality through the Uncertain and Reversal session (Fig. 4n). Notably, a much larger portion of dual-function neurons (20%, 98 out of 493 neurons, Fig. 4m) were recruited in the Reversal session compared to the Uncertain session, indicating an underlying neural mechanism for accelerated behavior switch. To estimate how it happened, we built a reinforcement learning model to estimate the trial-by-trial change of incentive value ($Q$) of 3 kHz tone updated with the feedback of prediction error ($\Delta r$). The learning rate ($\alpha$) in this modified state-action-reward-state-action (SARSA) model was also trial-by-trial modulated by the prediction error and the previous learning rate ($\alpha_t = \theta' \alpha_{t-1} + \theta |\Delta r_{t-1}|$). The model well predicted the change of 3 kHz-associated value and licking probability (Supplementary Fig. 10). The yielded learning rate was higher in the early Reversal session than that in the Uncertain session (Fig. 4o), similar to the 2-fold increase of dual-function neurons recruited in the Reversal session. One possible function of the dual-function neurons could be that each of them worked as a parallel computation unit for feedback-guided value iteration. Dynamic recruitment of them could regulate the converting of error feedback to value updating and the speed of decision switch, in order to achieve flexible, yet robust enough, feedback-guided value and decision update.

## Expected feedback prevents the net drift of value representation but not outcome-guided value iteration

In both the Stable and Re-stable sessions, the stimulus-reward association was fixed but opposite to each other. The behavior choice and the neuronal response to the same tone were completely reversed in the Re-stable session compared to the Stable session (Fig. 5a). The behavior performance and population response were steady through the early (33%, T1), middle (33%, T2) and late (33%, T3) phases in both sessions, except the go cue evoked response was slightly decreased in T2 in the Re-stable session (Fig. 5b–e). The population responses to the go stimulus correlated strongly in the three phases in the same session (Fig. 5f, left and middle), but became decorrelated between the two sessions (Fig. 5f, right). To detect unexpected outcome, the outcome monitoring and value updating function should be constantly carried out even when the outcomes turn out to be expected. Although the impact of experience history on the stimulus representation over the whole session was balanced by the lack of error feedback, the impact of the outcome on the immediate next few trials might still be traceable. To test this, we divided all go trials of Stable and Re-stable sessions into two groups by the outcome of the previous trial (post reward or post non-reward go trials, Fig. 5g). The go stimulus-evoked ACC responses were different when the animal experienced reward or non-reward outcomes in the previous trial (Fig. 5h). We used general linear models (GLM) to fit neural responses evoked by cue stimuli and assess the impact of the outcomes from the five most recent trials (Fig. 5i). In both the Stable and Re-stable session, the recent trials had influence on the go stimulus representation of the current trial as their coefficients were significantly different from 0 ($P < 0.05$, two-sided one-sample Wilcoxon test, Fig. 5j), and the experience history had a much stronger and longer influence on ACC neuronal activity in the Re-stable session (larger absolution value of the coefficients, Fig. 5j, right), suggesting that experiencing the drastic change in task rule might have a long-term impact on feedback-guided value iteration in ACC. Experiencing repetitive unexpected punishment and learning to associate the previous no-go cue to reward during reverse learning led to the investment of more neuronal resources to store and integrate outcome information even when the subject was able to reliably perform the current task, just in case unexpected incidents occurred again in the near future. Thus, the relatively long history of error experience was also able to trigger the dynamical recruitment of feedback-guided iteration units in the ACC network.

## Suppression of ACC activity delayed feedback-driven decision switch without affecting the correct choice of following established decision rules

Value representation, as the fundamental function for goal-directed behaviors, was well secured in the decision-making network, and perhaps, also redundantly presented in many other areas of the cortex[20]. Disrupting value representation would directly impair the decision-making process. Whereas, value iteration, as a higher-level function to optimize decisions, is crucial for switching to a new decision strategy rather than applying the established one. Disturbance of value iteration might block or slow down the strategy switching without too much interference to execute the established decision rule. To test whether the key function of ACC in the decision network is value iteration or representation, we expressed eNpHR in ACC excitatory neurons and suppressed their activity with 589 nm laser during the stimulus window when 3 kHz tone was presented in the Stable, Uncertain and Reversal sessions, respectively (Fig. 6a–c, f). When ACC activity was suppressed during the Stable session, animals were able to

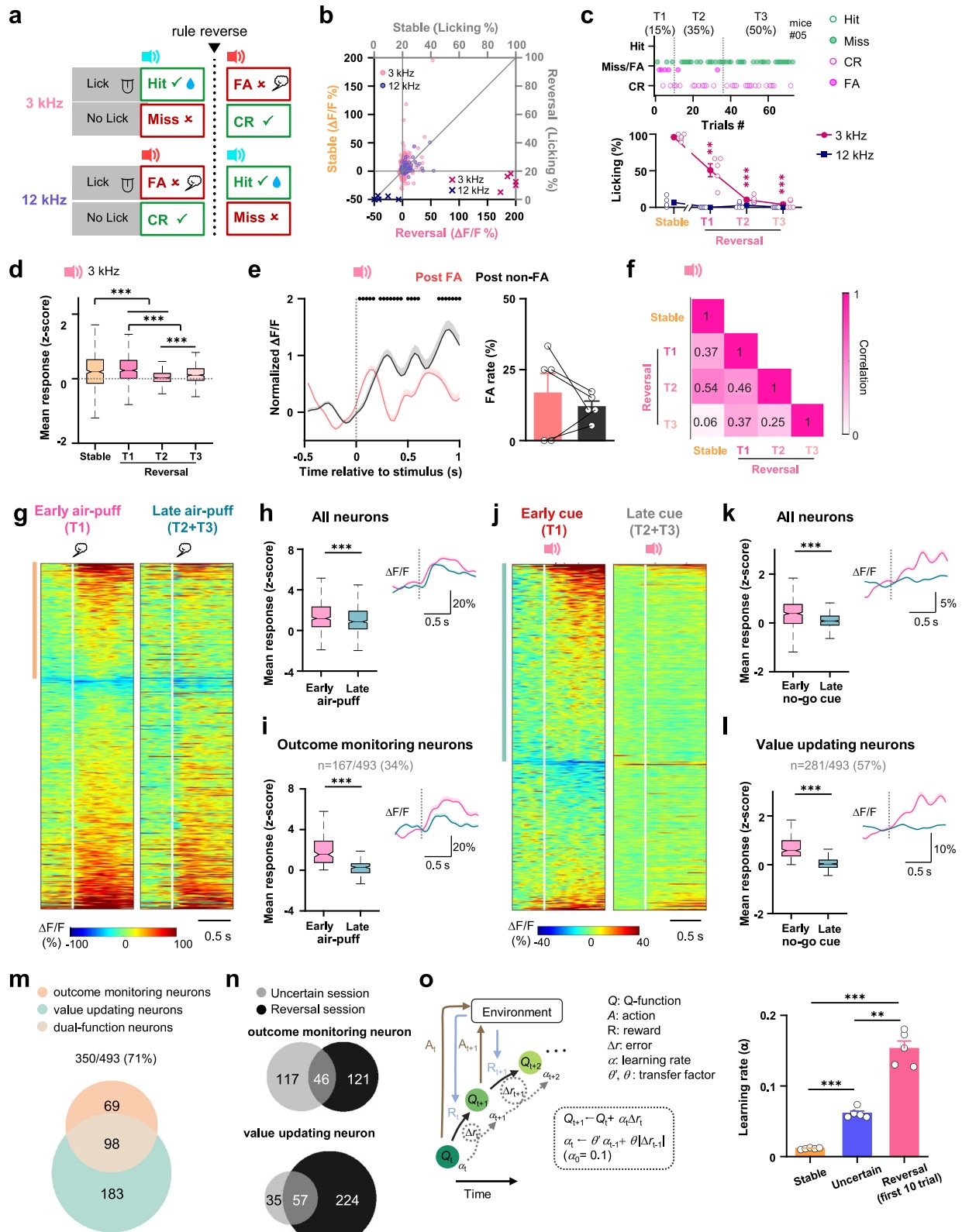

maintain the similar high Hit rate and discrimination performance (Fig. 6b). Whereas in the Uncertain session, suppressing ACC activity during stimulus window stopped the increase of miss trials compared to the mCherry group (Fig. 6c, d), and therefore blocked the decrease of discrimination performance induced by the unexpected outcomes (Fig. 6e), which could be well observed in the control group. Moreover, when the stimulus-reward contingencies were reversed, the control

animals stopped licking upon 3 kHz tone after a few times of punishment (Fig. 6f–h), but animals in the eNpHR group took a few more licks and air-puff punishment until they finally gave up the licking decision previously associated with 3 kHz tone (Supplementary Fig. 11). As a result, the eNpHR group exhibited significantly higher licking probability than the control group in the Reversal session. Similar effect was also observed when ACC activity was ontogenetically suppressed

**Fig. 4 | Reverse learning drives fast switch of decision and value representation.**
**a** Scheme showing stimulus-outcome contingency reversal. **b** Mean response to
3 kHz and 12 kHz tones of each neuron in the Stable and Reversal sessions. **c** Top:
Example behavior performance in the Reversal session. Bottom: licking probability
in the Stable session and the Reversal session. **d** Mean population response to 3 kHz
tone. **e** Left, average population responses to no-go stimulus following FA trials
(red, post FA) and following other trials (black, post non-FA). Right, the FA rate of
trials following FA or non-FA trials. **f** Correlation of 3 kHz responses. **g** Trial-
averaged neuronal responses to air puff in early and late periods. Air-puff evoked
response in the early and late periods in all neurons (**h**) and identified outcome
monitoring neurons (**i**). Inset: Population-averaged $Ca^{2+}$ response (mean ± s.e.m.).
**j**–**l** Similar to **g**–**i**, but analysis for the no-go cue evoked response. **m** Venn diagram
of the number of functional neurons in the Reversal session. **n** Venn diagram of the

number of functional neurons identified in the Uncertain and Reversal sessions.
Note that 56 neurons recorded in the Uncertain session could not be tracked in the
Reversal session. **o** Left: schematic of the SARSA model. Right: average learning rate
of the value iteration of the 3 kHz tone. **$P < 10^{-2}$, ***$P < 10^{-3}$. Data analyzed by
(**c**, $n = 5$ mice) two-sided one-way repeated measures ANOVA with post-hoc Dun-
nett's comparisons (comparing to Stable session), (**d**, $n = 493$ neurons from 5 mice)
two-sided Friedman test with post-hoc Bonferroni comparisons, (**e**, $n = 5$ mice) two-
sided paired $t$-test, (**h**, **k**; $n = 493$ neurons; **i**; $n = 167$ neurons; **l**; $n = 281$ neurons, from
5 mice) two-sided Wilcoxon signed-rank test, or (**o**; $n = 5$ mice) two-sided one-way
repeated measures ANOVA with post-hoc Tukey's comparisons. Data are presented
as (**c**, **e**, **o**) mean ± s.e.m. or (**d**, **h**, **i**, **k**, **l**) box plots (center line, median; box limits,
upper and lower quartiles; whiskers, 1.5 × interquartile range). Statistical details are
presented in Supplementary Table 1. Source data are provided as a Source data file.

during the response window when unexpected outcomes were
experienced in the RO, UR and FA trials (Fig. 6i–n). It proved that the
activity of ACC neurons was crucial for switching to a new decision
driven by unexpected feedback, rather than correctly applying the
previously established strategy.

## Discussion

In the natural environment, the cues for food opportunities or pre-
dator threats can be present in various forms at any time, and it is
essential for the animal to quickly learn the cues and be flexible enough
to adjust the decision-making strategy accordingly. Such flexible
adjustment of decision-making is the hallmark of intelligence and the
fundament for higher brain functions. Decision-making is based on the
established value representation of the input stimulus, and the flexible
adjustment of decision strategy relies on the modification of stimulus
value triggered by the unexpected outcomes. It requires two func-
tional elements, including an outcome monitoring system to detect
error ($\Delta r$) and a value updating system to integrate the previous
feedback message ($\alpha \Delta r$, with $\alpha$ as the learning rate determining the
impact of feedback information) to generate the new value repre-
sentation to guide behavior decision[2,21,22] (Fig. 1a). Several brain areas,
including ACC[1,9,23], the medial prefrontal cortex[24–27] and striatum[28–30],
have been reported able to detect errors and encode value informa-
tion. For example, the orbitofrontal cortex encoded reward and pre-
diction related information in goal-directed behaviors and the
response of OFC neurons increased after reverse learning[31,32]. Similar
upregulated response and selectivity after reverse learning (Re-stable
session) was also observed in ACC (Supplementary Fig. 2b and 9f),
indicating that the ACC and other frontal cortical areas including OFC,
formed a larger network providing necessary information to the
decision-making process.

A key question here is how the error signal is converted to the
renewed value in the next-run decision process. A group of ACC neu-
rons can simultaneously encode both the outcome feedback and the
renewed value representation (40/549 neurons in the Uncertain ses-
sion and 98/493 neurons in the Reversal session, Figs. 3g and 4m,
respectively). They were strongly activated when experiencing reward
omission or punishment, and downregulated their response to the
stimulus. The response of the dual-function neurons to the 3 kHz tone
and the air puff in the early trials of the Reversal session correlated well
with the Q-value and prediction error estimated by the SARSA model
(Supplementary Fig. 10). The same neuron not only gathered the
feedback information, but also retained it until the next-run stimulus
started, and integrated it to value representation of the stimulus.
Therefore, each dual-function neuron worked as an elementary
storage-computation unit capable of temporally cumulating error
signals to shift the internal value representation and decision strategy.

The next question is how the error feedback efficiently and
appropriately determines the speed of the value and decision switch.
An ideal system should be resilient to single incidents but flexible
enough for rapid switches when experiencing repeated error events.

When the outcome always confirms the expectation, the impact of
feedback stays low, and the decision strategy is robust against an
occasional error event. When the error signal is repetitively detected,
error not only directly contributes to the value updating, but also
increases the learning rate ($\alpha$) which determines the impact of the
error signal, forming a non-linear error accumulation of system
($(\theta' \alpha + \theta |\Delta r|)\Delta r$) capable of accelerating the decision adaptation. The
key agent in the system is the dual-function (outcome monitoring and
value updating) neurons, which works as the parallel computation
units for feedback-guided value iteration. The number of recruited
dual-function neurons is proportion to the learning rate, as the learn-
ing rate and dual-function neurons both doubled in the Reversal ses-
sion compared to the Uncertain session (Figs. 3g, 4n and 4o). The error
signal triggers the recruitment of dual-function neurons, which
determines the neural resource to compute the value iteration, as well
as the impact of the feedback information. If the outcomes of the next
few events are the same as expected, the recruited dual-function
neurons would be released, the learning rate decreased, and the
established decision strategy would be resilient to the error event. As
in the Stable session the learning rate reduced to near zero (Fig. 4o),
the value representation of the stimulus was stable (Fig. 5d), and the
decision was robust even when the occasional error signal (FA trials)
occurred (Fig. 5c). On the other hand, when strong error events were
consecutively experienced, such as in the Reversal session, it triggered
rapid recruitment of a much larger number of dual-function neurons
(98/493 ACC neurons, Fig. 4m), enhancing the impact of the strong
punishment signal on the value updating, and efficiently stopped the
licking response after a few FA trials.

Decision flexibility is a critical element of complex brain function
and is essential for survival and adaptation in ever-changing environ-
ments. In the decision-making network, many brain areas are able to
encode the value information, whereas ACC neurons provide neuro-
logical substrates for feedback-guided value iteration, which is crucial
for decision adjustment rather than making the correct decision with
an established strategy (Fig. 6). It directly proved that ACC as a core
region of higher-level cognitive function to optimize the decision-
making process. It conveys updated value information to
hippocampus[33–35] and amygdala[36,37] for the formation and retrieval of
value-associated memories, and also shares intense reciprocal con-
nections with other cortical areas where sensory and motor informa-
tion is analyzed and extracted[8,38–40]. The ACC collects and converts the
sensory and feedback information to update value information, pro-
vides adjustment to optimize the decision-making process[41]. The
updated value information is also very important top-down regulatory
signals, which was sent back directly or indirectly to the sensory and
motor cortices[13,39,42] to guide the behavior-relevant sensory processing
and memory formation. Such as in the primary visual cortex, the per-
ceptual learning enhanced visual response and selectivity might be
mediated by the precise top-down modulation from ACC[42].

To summarize, we found individual ACC neurons were able to
convert the feedback information to the updated value representation

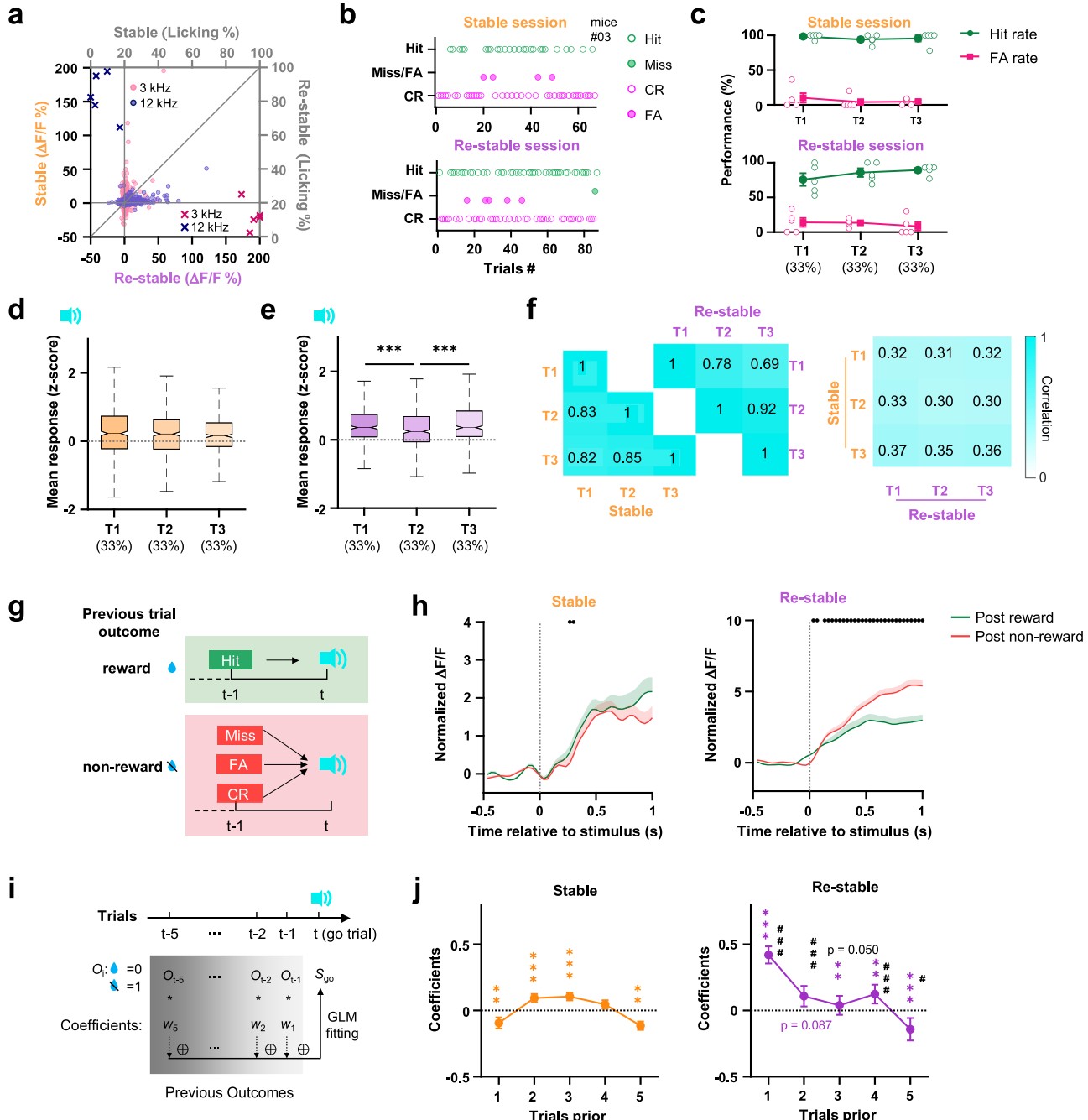

**Fig. 5 | Feedback-driven iteration of value representation is continuously performed even in the absence of error feedback. a** Scatter plots of the mean stimulus response of each neuron in the Stable and Re-stable sessions (*n* = 493 cells from 5 mice). **b** Behavior performance of an example animal in the Stable and Re-stable sessions. **c** Behavior performance in T1-T3 phases in the Stable and Re-stable sessions (*n* = 5 mice, two-sided one-way repeated measures ANOVA with post hoc Tukey's multiple comparisons). Mean population response to go stimulus in T1-T3 phases in the Stable session (**d**) and the Re-stable session (**e**). *n* = 493 neurons from 5 mice, two-sided Friedman test with post hoc Bonferroni multiple comparisons. **f** The correlation of go-cue evoked population activity between the three phases in the Stable (left) and Re-stable (middle) sessions, and between those two sessions (right). **g** The scheme of go trials grouped by the reward outcome of the previous trial. **h** Average population responses to go stimulus following reward outcomes (green, post reward trials) and non-reward outcomes (red, post non-reward trials) during the Stable and Re-stable sessions, respectively. Black dots indicate the time segments when traces are different (*P* < 0.05, two-sided Wilcoxon rank-sum test). **i** Scheme of constructing GLM. **j** Regression coefficients of outcome history of previous trial contributing to the neural response to the go stimulus in the Stable and Re-stable session (*n* = 493 cells from 5 mice). The symbol * denotes a significant difference compared to 0 (two-sided one-sample Wilcoxon signed-rank test). The symbol # denotes the significant difference of coefficients of the previous trial between the Stable and Re-stable session (two-sided Wilcoxon rank-sum test). \*\**P* < 10⁻², \*\*\**P* < 10⁻³, #*P* < 0.05, ##*P* < 10⁻², ###*P* < 10⁻³. Data (**c**, **h**, **j**) are presented as mean ± s.e.m. or (**d**) box plots (center line, median; box limits, upper and lower quartiles; whiskers, 1.5 × interquartile range). Statistical details are presented in Supplementary Table 1. Source data are provided as a Source data file.

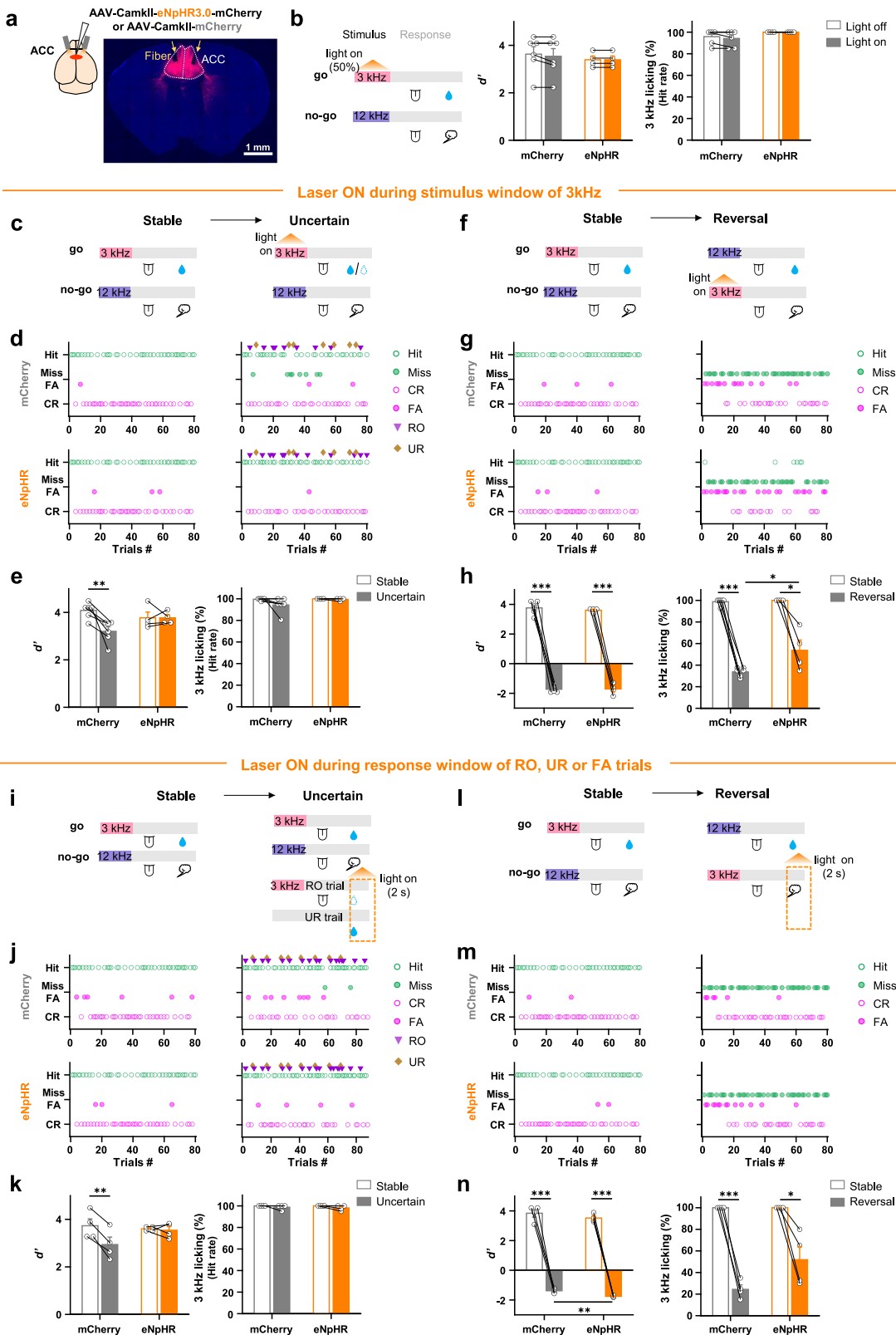

of the decision-relevant stimulus. Those dual-function neurons work as the elementary units of value iteration to determine the learning rate of value iteration and decision switch. The dynamic recruitment of them formed a non-linear feedback-guided value iteration system to ensure the change of decision strategy is appropriate and well-timed. These results revealed the neural substrates critical to the iterative process of optimizing decision strategy, and shed light on the further

understanding of the hierarchical function and structure of the decision-making network.

## Methods

### Animals

C57BL/6 J male mice (2–4 months old, 20–30 g, Vital River) were used in this study. All procedures were performed following protocols

**Fig. 6 | Optogenetic suppression of ACC impairs decision switching without disrupting the implementation of established decision rules. a** Top left: scheme of optogenetic suppression of the ACC neurons. Right: representative images of eNpHR expression and the position of the implanted fiber above ACC ($n = 4$ mice). **b** Suppressing ACC activity in the Stable session does not harm the task performance. 589 nm laser suppression is delivered through bilateral optic fibers during the 3 kHz stimulus window in randomly selected 50% of go trials (left). There was no significant change in behavior performance for $d'$ (middle) and 3 kHz licking probability (Hit rate, right) during ACC inhibition. **c–h** Suppressing ACC activity during 3 kHz stimulus window in the Uncertain and Reversal sessions blocked decision switch. **c** 589 nm laser is delivered during the 3 kHz stimulus window in the Uncertain session but not in the Stable session. **d** Performance of a representative animal in the mCherry (top) and eNpHR mice (bottom) in Stable (left) and Uncertain (right) sessions. **e** Behavior performance for $d'$ (left) and

3 kHz licking probability (Hit rate). **f–h** Suppressing ACC activity in the Reversal session slowed down the decision switch. Similar to **c–e** except optogenetic suppression was delivered during the 3 kHz stimulus in the Reversal session. **i–n** Suppressing ACC activity while experiencing the unexpected outcome impairs decision switching. Similar to **c–h**, except 589 nm laser with a duration of 2 s is delivered after the reward omission (RO trial) and un-cued reward (UR trial) was triggered in Uncertain session, or after the air-puff was triggered in Reversal session. *$P < 0.05$, **$P < 10^{-2}$, ***$P < 10^{-3}$. Data analyzed by (**b, e, h:** $n = 6/4$ mice for mCherry/eNpHR group; l, o: $n = 4/4$ mice for mCherry/eNpHR group) two-sided two-way repeated measures ANOVA, within-group comparisons: one-sided paired t-test with post hoc Bonferroni multiple comparisons; between-group comparisons: one-sided unpaired t-test with post hoc Bonferroni multiple comparisons. Data are presented as mean ± s.e.m. Statistical details are presented in Supplementary Table 1. Source data are provided as a Source data file.

---

approved by the Animal Care and Use Committee at Huazhong University of Science and Technology. All surgery was performed under full anesthesia, and every effort was made to minimize animal suffering. The animals were singly housed under a 12 h light/dark cycle and all experiments were performed during the light cycle. Standard mouse chow and water were provided *ad libitum* unless specified otherwise.

## Surgery

For surgical procedures, all mice were subcutaneously (s.c.) injected with dexamethasone (5 mg/kg) 1 day before surgery to prevent cerebral edema. Mice were anesthetized with an intraperitoneal (i.p.) injection of 1.25% tribromoethanol (0.3 ml/10 g) and local injections of 1% lidocaine (0.02 ml/animal; s.c.) were given to provide analgesia before opening the scalp. While the mice were under deep anesthesia, they were placed in a stereotaxic frame (RWD Life Science), and their core body temperature was kept at 37 °C with an electric feedback-controlled heating pad.

For in vivo imaging of ACC neurons, a cranial window was implanted in all mice before behavioral training as described before[43]. rAAV$_{2/R}$-mCherry (50 nL, $10^{12}$ VP ml$^{-1}$, BrainVTA) were injected into the retrosplenial cortex (RSP, AP: −2.7 mm; ML: −1 mm; DV: −0.5 mm; Allen Mouse Brain Atlas) at a rate of 5 nL min$^{-1}$ via a calibrated glass pipette (5 μL microcapillary tube; Sigma-Aldrich) connected to a pneumatic picopump (SYS-PV830, WPI). Circular craniotomy (3 mm in diameter, centered at 0.3 mm lateral and 1 mm anterior of Bregma) was performed. rAAV$_{2/9}$-CamKII-GCaMP6f (50 nL, $10^{10}$ VP ml$^{-1}$, BrainVTA) was injected into the ACC (AP: +1.0 mm; ML: −0.3 mm; DV: −1.2 mm; Allen Mouse Brain Atlas). The craniotomy was sealed with a 3 mm glass coverslip and cyanoacrylate glue (Pattex), and a titanium alloy head-plate was attached to the skull with C&B Super-Bond (BearDayton).

For optogenetic manipulation, rAAV$_{2/9}$-CamkII-eNpHR3.0-mCherry (50 nL, $10^{12}$ VP ml$^{-1}$, BrainVTA) or rAAV$_{2/9}$-CamkII-mCherry (50 nL, $10^{12}$ VP ml$^{-1}$, BrainVTA) was bilaterally injected into the ACC (AP: +1.0 mm; ML: ± 0.3 mm; DV: −1.2 mm). The pipette was retracted 10 min after injection, and optical fibers (200 μm O.D., 0.37 NA; Inper, China) were implanted bilaterally over the ACC (AP: +1.0 mm; ML: ± 0.6 mm; DV: −1.1 mm; angle; ±10°). The optic fiber and head-plate were secured using C&B Super-Bond (BearDayton).

The animals were kept in a 37 °C post-surgery recovery chamber for least 3 h before returning to the home cage. Tolfenamic acid (0.5 mg/kg, Tolfedine, Vetoquinol) and enrofloxacin (0.5 mg/kg, Baytril, Bayer) were given (s.c.) daily for at least five days to alleviate pain and inflammation.

## Histology

After completion of all recordings, the mice were deeply anesthetized and transcardially perfused with 4% paraformaldehyde (PFA) in phosphate-buffered saline (PBS). Brains were post-fixed in 4% PFA in PBS at 4 °C overnight, followed by dehydration in 30% sucrose in PBS

for 48 h. Coronal sections of 30-μm thickness were performed using a Cryostat microtome (Leica CM1950) and imaged using a slide scanning system (Olympus VS120) with a 10× objective (NA 0.4).

## Behavioral training

After at least one week of post-surgery recovery, the mice were handled for 10 min to 1 h each day for at least 3 days before the initial training. The mice were food-restricted to reach 80–90% of their free-feeding body weight and trained on the auditory go/no-go discrimination task. The behavioral experiments were controlled and recorded by the Arduino-based platform and ArControl software[44].

The initial training process consisted of three phases: *habituation*, *association* and *discrimination*. In *habituation* phase (~ 5 min), mice were first trained to trigger 10% sucrose water reward (5 μL, Thermo Fisher Scientific) reliably by licking the spout. In the subsequent *association* phase, only the go cue (3 kHz, 70 dB) was presented for 1 s, followed by a 2-s response window in which the mice could be rewarded with sucrose solution by licking the spout. To prevent mice from licking continuously, the stimulus did not start until the mice stopped licking for at least 4–6 s. The *association* phase took about 3 days and mice learned to suppress impulsive licking and lick only during the stimulus and response window. Then mice were trained to discriminate between two different tones (go cue: 3 kHz, 70 dB; no-go cue: 12 kHz, 70 dB) in *discrimination* phase, which was defined as day 1 in the task training (Supplementary Fig. 1a). The go and no-go cues were presented randomly with a 4–6 s inter-trial interval (ITI). The reward was only delivered for licks in the response window following the go cue (Hit). Incorrect licking in response to the no-go cue (FA: false alarms) was punished with a puff of air (0.2 s, 20 psi, through a flattened 25 g needle) towards the whiskers. Licking during the stimulus window was not rewarded or punished. Neither reward nor punishment was given when mice withheld licking for the go cue (miss) or no-go cue (CR: correct rejections). The mice typically learned to perform the normal go/no-go task within 3–7 days (150–200 trials per day). The animals were considered proficient if they achieved stable high behavioral performance of the go/no-go task ($d' \geq 1.5$) for at least three days. For Reversal training, the stimulus-outcome association was switched ('rule switch', go cue: 12 kHz, 70 dB; no-go cue: 3 kHz, 70 dB). It took 5–7 days of reverse learning for the animal to reach high behavioral performance ($d' \geq 1.5$ for at least three days), and the Re-stable session was recorded.

## Two-photon calcium imaging

Two-photon calcium imaging was performed using a resonant-galvo scanning two-photon microscope (Scientifica; 30 fps; 512 × 512 pixels) through a 16×/0.8NA water-immersion objective (Nikon) with excitation by a mode-locked Ti: sapphire laser at 920 nm (30–50 mW under the objective, Mai Tai eHP DeepSee, Spectra-Physics, USA). Imaging region of ~ 250 × 250 μm² was located lateral to the superior sagittal sinus at a depth of 250– 400 μm below the pial, corresponding to the

posterior dorsal ACC. Images were acquired by ScanImage software (Vidrio Technologies). For repeated imaging across sessions, the imaged region was identified by the vascular landmarks.

The mice were imaged in four typical sessions. After the mice achieved stable high performance of the normal go/no-go task, they were imaged for the Stable session, which was composed of 50%/50% randomly interleaved go/no-go trials. The Uncertain session consisted of 40% of go trials, 40% of no-go trials, 10% of trials with go cues followed by no reward delivery with correct licks, and 10% of trials with rewards delivered with no stimulus cue. The Reversal session consisted of 50%/50% go/no-go trials with switched stimulus-outcome associations. The Re-stable session was the same as the Reversal session except it was a few days of training after the Reversal session when the mice had achieved steady high performance ($d' \geq 1.5$) for at least three days.

### Behavioral performance measurement

We quantified the task performance by measuring the discriminability index d-prime ($d'$) as:

$$d' = norminv \text{ (Hit rate)} - norminv \text{(FA rate)} \quad (1)$$

where $norminv$ is the inverse of the cumulative Gaussian distribution; Hit rate = the number of Hit trials / the total number of go trials. FA rate = the number of FA trials / the total number of no-go trials. We set the learning threshold as $d' = 1.5$[31]. In the Uncertain session, the correct licking in the RO trials was also considered as Hit because the animal made the correct licking choice upon the go cue.

### Optogenetic suppression of ACC activity

For optogenetic experiments, animals were initially trained to perform the go/no-go discrimination task with randomly presented go and no-go cues (go cue: 3 kHz; no-go cue: 12 kHz). After the mice had reached a stable high performance ($d' \geq 1.5$ for at least three days), optogenetic suppression was performed as follows: (i) in the Stable-only block (50%/50% randomized go/no-go task; at least 80 trials/session), the 589 nm laser (10–15 mW) was on in the stimulus window in half of go trials (light-on) in the Stable session, while the other half were controls (light-off). The light-on and light-off trials were randomly interleaved; (ii) in the Stable-Uncertain block, the Stable session was followed by the Uncertain session (40% of go trials, 40% of no-go trials, 10% of reward omission trials, and 10% of un-cued reward trials; 80 trials/session). 589 nm laser was delivered in the stimulus window of all go trials (duration: 1 s), or delivered after the reward omission (RO trial) and un-cued reward (UR trial) was triggered (duration: 2 s) during the Uncertain session; (iii) and in the Stable-Reversal block, the Stable session was followed by the Reversal session (50%/50% go/no-go trials with switched stimulus-outcome associations; 80 trials/session). 589 nm laser was delivered in the stimulus window of all no-go trials (3 kHz tone, duration: 1 s), or delivered after the air-puff (FA trial) was triggered (duration: 2 s) during the Reversal session (Fig. 6).

### Calcium imaging analysis

Image stacks of two-photon calcium microscopy were first motion-corrected using the EZcalcium toolbox[45] in MATLAB (MathWorks). Regions of interest (ROI) corresponding to individual neurons were manually selected and fluorescence time courses (F) were then extracted using ImageJ software (US National Institutes of Health) as the mean pixel value for each ROI. In addition, the fluorescence signals were mapped to 0-1 and low-pass filtered at 5 Hz. ΔF/F was calculated as:

$$\Delta F/F = (F - F_0)/F_0 \quad (2)$$

where $F_0$ is the mean between the 25th percentile and 75th percentile of the baseline fluorescence signal over a 2 s period before each stimulus onset. The normalized ΔF/F was calculated by subtracting the mean of the baseline (0.5 s before stimulus onset or reward/reward omission/punishment delivery) and dividing by its standard deviation (s.d.).

The neurons were considered responsive only if their activity (ΔF/F) during the stimulus windows was significantly different from the baseline period (2 s before stimulus onset, $P < 0.05$, Wilcoxon signed-rank test)[46]. Neurons with significant response were further classified as activated or suppressed based on their increased or decreased response relative to baseline, respectively (Supplementary Fig. 2c). The onset time of response was determined by the first time point (smoothed with a 3-frame window) when the stimulus-evoked response was significantly different from the baseline ($P < 0.01$, two-sided Wilcoxon rank-sum test, Supplementary Fig. 3d)[47].

The correlation of population activity was calculated as the Pearson's correlation coefficients of the trial-averaged response of all the recorded neurons to the same stimulus in each phase (Figs. 2f, 4f, 5f, and Supplementary Fig. 4j, 8d).

### Identification of outcome monitoring and value updating neurons

**Uncertain session.** The mean response of each neuron within a 1-s window after the first lick in the response window following the go cue was calculated as the response to the reward delivery in Hit trials (~ 23–40 trials/animal) or the reward omission in RO trials (~ 6–13 trials/animal). Outcome monitoring neurons were defined as those with mean responses in the RO trials higher than that in Hit trials by at least 2 folds (R1-R2 > |R2|). (Fig. 3a–c). The Uncertain session was separated into the early (first 67% of trials) and late (last 33% of trials) periods. The mean response of each neuron to the go stimulus (within the 1-s window after stimulus onset) was compared. Value updating neurons were defined as those neurons whose go stimulus evoked mean response in the early period was higher than that in the late period by at least 2 folds. (Fig. 3d–f).

**Reversal session.** The Reversal session was separated into the early (first 15% of trials) and late (last 85% of trials) periods. The mean responses to the air-puff (within a 1-s window after air-puff delivery) in the early and late periods were compared. Neurons with mean air-puff evoked responses that were at least twofold higher in the early period than in the late period were defined as outcome monitoring neurons (Fig. 4g–i). Neurons with mean no-go cue evoked responses (during 1-s window after stimulus onset) in the early period that were at least twofold higher than in the late period were defined as value updating neurons (Fig. 4j–l).

### Principal component analysis of population activity

The trial-averaged population response to the go and no-go stimulus (z-score of ΔF/F from −0.5 s to 1 s relative to the onset of stimulus) was subjected to principal component analysis (PCA)[48]. The top three PC components captured >66% of the variation of population activity in all sessions (Fig. 1h). The length of activity trajectory was calculated as the sum of Euclidean distances in the principal component space (PC1-3) in the 1-s window during stimulus presentation. Distance of activity trajectories between the go and no-go cue evoked responses was calculated as the sum of Euclidean distance between the pairwise time points of the two trajectories during the stimulus window. For statistical comparisons, the trajectory distance of session was sampled by a bootstrapping approach ($n = 5,000$, a random set of $N = 40$ neurons from the population for every step, Fig. 1j, k). The mean response of each cell to the go and no-go stimulus (during the stimulus window) was subjected to PCA. The mean activity of the whole population in

each session was projected to the space of the top two PCs, which captured > 89% of the variation (Fig. 1l).

## Constructing classifiers and decoding population activity

The decoding results were obtained with linear support vector machine classifiers (SVM) implemented using a cross-validation procedure[49,50]. All procedures were repeated 500 times, and the zero-one loss measure was used to report the accuracy of the decoding procedure.

**Decoding stimulus identity.** For cross-phase decoding in Fig. 2, the dataset included the ΔF/F response in the stimulus window (0-1 s) of all go and no-go trials during four stages, including Stable session and T1-T3 phases (first, middle, and last 33% trials) of the Uncertain session. The classifier was trained with a 3-fold leave-one-split-out cross-validation procedure in which two-thirds of trials in the same session or phase were randomly selected for training and the remaining trials were used for testing (Fig. 2g). For the cross-phase decoding test, the classifier trained by each group of data (Classifier I - IV) was used to decode neural activity of all the trials in the test phase (Fig. 2h, i and Supplementary Fig. 4k).

**Outcome and session periods decoding.** We compared the decoding accuracy of the whole population ($n = 549$ neurons) and the sub-population without the dual-function neurons ($n = 509$ neurons). To eliminate the influence of the different numbers of neurons in those two groups, the decoding test was performed with the same number of neurons ($n = 200$) randomly selected from each cohort. A 5-fold cross-validation was performed, and each decoding analysis was repeated 1000 times for the Bootstrap test. The z-score of ΔF/F was used for decoding. For outcome categories decoding, the mean response during the outcome window (0-1 s after reward delivery or omission) in Hit and RO trials. Similarly, for session period decoding, we used the mean response of neurons during the stimulus window (0-1 s after stimulus onset) in go trials of early and late periods (Fig. 3h).

## Generalized linear model

To quantify the contribution of task variables to neural activity, we fitted a generalized linear model (GLM) for go trial during the Uncertain session[51]. In brief, we used four task variables (stimulus, cumulative history, previous outcome, and licking rate) to fit the per trial ΔF/F signal during stimulus window (0–1 s after the onset of the go cue). The stimulus variable was set to '1' at all time points. The cumulative history variable ranges from $0 – 1$, and represent the percentage of the total number of previously experienced unexpected trials (RO and UR). The previous outcome history variable was set to '1' if the previous trial was an unexpected outcome, including RO and UR trial or '0' if the previous trial was a normal outcome, respectively. The relative contribution of each task variable was quantified by calculating how much the explained variance decreased when each variable was removed from the model compared to using all of the variables (5-fold cross-validation was used in all cases).

Another GLM was conducted to account for the influence of the recent history of trial outcomes on the neuronal response for go cue. We fit the normalized responses of all neurons in Hit trials in the go/no-go task with the following linear models:

$$f(t) = \alpha + w_1 * O_{(t-1)} + w_2 * O_{(t-2)} + w_3 * O_{(t-3)} + w_4 * O_{(t-4)} + w_5 * O_{(t-5)} \quad (3)$$

where f(t) is the mean stimulus response (0–1 s after go-cue onset, z-score of ΔF/F from −0.5 s to 1 s relative to the onset of stimulus) in trial t; $O$ is the outcome type (the non-reward and reward were assigned as 1 and 0, respectively) of recent trials; α represents the basic sensory response of the stimulus (as the intercept of the GLM), and $w_i$ is the impact of the outcome of trial t-i (as the regression

coefficient weight for $O_{(t-i)}$). These models can capture variability in the response caused by the outcome history of previous trials (Fig. 5i, j).

## SARSA model

A modified State-action-reward-state-action (SARSA) model was applied to estimate the trial-by-trial incentive value of the 3 kHz tone (Q) and perdition errors (Δr), with the choice of action as lick or no-lick ($A_{lick}$ or $A_{nolick}$, respectively).

$$Q(S,A) \leftarrow Q(S,A) + \alpha \Delta r = Q(S,A) + \alpha(R - Q(S,A)) \quad (4)$$

The learning rate (α) was also updated trial-by-trial with a transfer rate of $\theta'$ and $\theta$ for the learning rate in the previous trial and the prediction error.

$$Q(A_{j+1}) \leftarrow Q(A_j) + \alpha_j \Delta r^j = Q(A_j) + (\theta' \alpha_{j-1} + \theta |\Delta r^{j-1}|) * \Delta r^j \quad (5)$$

The reward of the action (R) was set to 1 (Hit), 0.1 (Miss) and −0.1 (RO) in the Stable and Uncertain session, and −2 (FA) and 0.1 (CR) in the Reversal sessions, respectively. The initial state was set as $\alpha_0 = 0.1$, $Q_0^{lick} = 0.98$ and $Q_0^{nolick} = 0.02$, the transfer rate was set as $\theta' = 0.6$ and $\theta = 0.08$ for all sessions.

Simulated task performance was repeated for 3000 times with the same action-outcome rule in each session. The action sequence was generated based on the trial-by-trial licking probability (P), which was estimated by the value of the two-action choice ($Q_i = (\{Q_n^{lick}\}, \{Q_m^{nolick}\})^T$), according to the QP transformation function with a linear proportional scaling index ($\beta$) derived from the experimental performance ($\beta = 2.3$ for the Stable and Uncertain sessions and $\beta = 1.68$ in the Reversal session).

$$P = f(Q_i, \beta) = \frac{e^{\beta Q_i}}{\sum_{j=1}^{n} e^{\beta Q_j}} \quad (6)$$

## Selectivity index (SI)

To quantify the selectivity of neural responses to the go and no-go cues, we computed a response selectivity index (SI) for individual cells from the difference of the mean response in the 1-s window of the go and no-go cues ($\overline{R_{go}}$ and $\overline{R_{no-go}}$, respectively), divided by the pooled standard deviation of the responses[52]:

$$SI = (\overline{R_{go}} - \overline{R_{no-go}})/\sigma_p; \quad (7)$$

where

$$\sigma_p = \sum_{i=1}^{k=2}(n_i - 1)s_i^2 / \sum_{i=1}^{k}(n_i - 1) \quad (8)$$

and $n_i$ is the number of trials in condition $i$ for $k$ conditions. Therefore, the positive values indicate a preference for the go cue and the negative values for a preference for the no-go cue. To test whether the neuron exhibited significant preference for the go or no-go cue, a permutation test was performed, in which SI was calculated when the trial type was shuffled for 1000 repeats. A neuron was considered to have a significant selectivity if its actual SI exceeded 95% of the shuffled results ($P < 0.05$, Supplementary Fig. 9f, g).

## Statistics

Statistical analyses were performed with scripts written in MATLAB (2020a, MathWorks) and GraphPad Prism 9 (GraphPad). $P < 0.05$ was considered statistically significant for all data. All statistical analyses

are described in the Supplementary Table 1 (Statistic Table) and Figure legends.

## Reporting summary

Further information on research design is available in the Nature Portfolio Reporting Summary linked to this article.

## Data availability

The data that support the findings of this study are available in Figshare (https://figshare.com/s/eee94309c2b4297357fb). Source data are provided with this paper.

## Code availability

Custom code to analyze the data is publicly available (https://github.com/ChenWq2023/ACC and https://doi.org/10.5281/zenodo.12177591).

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

## Acknowledgements

This work was supported by National Natural Science Foundation of China (32271082 to Y.H. and 52005191 to J.L.), Junior Thousand Talents Program of China, Major Scientific Research Facility Project of Jiangsu Province (BM2022010 to Y.H.), Natural Science Fund for Distinguished Young Scholars of Hubei Province (2022CFA067 to Y.H.), Interdisciplinary Research Program of HUST (2024JCYJ007 to Y.H.), and State Key Laboratory of Intelligent Manufacturing Equipment and Technology (IMETZZ2024008 to J.L.). We are very grateful to Dr. Yuanyuan Mi for the critical suggestions and the help about modeling. We very much appreciate to the valuable advice from Dr. Haohong Li. We thank very much to the helpful discussion with Dr. Man Jiang, Dr. Zheng Guo, Dr. Luoying Zhang, Dr. Shangbang Gao, Dr. Bo Xiong, Dr. Yan Zhang and Dr. Hongbo Jia. Thanks for the support of image acquisition from Innovation and Research Center in School of Basic Medicine (HUST) and Medical Subcenter of HUST Analytical & Testing Center.

## Author contributions

Y.H. and W.C. designed the project, W.C. carried out the animal experiments, Y.H., W.C., J.L. and Q.W. analyzed the data, and Y.H., W.C. and J.L. wrote the manuscript.

## Competing interests

The authors declare no competing interests.
