## [Peer Review File · Nature Communications]

Anterior cingulate cortex provides the neural substrates for feedback-driven iteration of decision and value representationREVIEWER COMMENTS

Reviewer #1 (Remarks to the Author):

In this study, authors conducted longitudinal 2p imaging of ACC neurons when mice perform go-nogo task in the head-fix condition. Importantly, they examine how the neuronal activity and behavior change upon choice-outcome relationship becomes either uncertain or switched. They found that significant number of ACC neurons show increased activity when reward was unexpectedly omitted, and also show reduced activation to cues in the next-run trials when the outcome was unexpected. This pattern on reduced activity to cue gradually get accelerated overtime. Behavior changes correlate with the ACC activity changes by exhibiting gradual shift of decision strategy when experiencing repetitive unexpected outcomes. Authors also conducted optogenetic manipulation to suppress ACC activity during stimulus cue period and surprisingly showed that it slowed down the decision switching. Longitudinal imaging data presented are impressive combining decoding approach and informative to advance our understanding of how ACC integrate feedback information to potentially guide behavioral changes. However, the key causal optogenetic study does not well support the model driven from the imaging data, and are disconnected by missing key experiments. Better integration of imaging data and optogenetic causal manipulation is essential to support the main conclusion of the study.

Major points:

1: A major concern of this manuscript is a poor integration of the imaging data (Fig. 1-5) and the optogenetic manipulation study in Fig.6. While it is interesting that the optogenetic suppression of ACC neurons during cue period prevented the behavioral adjustment (Fig6e), the findings appear to be inconsistent to the imaging data showing reduced cue activity in uncertain condition (Fig. 2e, Fig3e). It looks to me that the findings of optogenetic manipulation do not support authors' working model. More thorough interpretation of the data and ideally bidirectional manipulation of cue period will help better understand the causal role of ACC neurons' activity changes in response to cues.

2: Related to the above point, it is unclear why authors chose to manipulate ACC neurons during auditory cue phase, but not during outcome phase. As imaging data suggests the importance of outcome feedback (Fig.3a-c, Fig4g-i). I think it is essential to examine the impact of outcome feedback period in the optogenetic manipulation study to support the authors' main conclusion.

Other points:

3: Please describe the quantitative criteria to define "outcome monitoring neurons" and "valence updating neurons" in Fig.3 and 4.

4: Fig.2e: It would be informative if authors describe how the post unexpected outcome activity changes as a function of the time. It would be helpful if they can show the break down with T1, T2, T3 as in Fig2d.

5: Similarly, Fig 3 bc can be break out to T1-3 to see if there is any time dependent changes.

6: It is helpful if authors document in which cortical layers the imaged neurons were located.

Reviewer #2 (Remarks to the Author):

Chen and Han studied an important question on how feedback renews and updates the internal value representation in the brain during adaptive behaviour. The authors importantly focused on a prefrontal brain structure, the anterior cingulate cortex (ACC), and studied whether and how stimulus-action remapping during behaviour is encoded in the ACC. They performed longitudinal Ca²⁺ imaging from populations of neurons in the ACC during a Go/No-Go task and found a subpopulation of ACC neurons integrated the outcome to value transformations in subsequent trials.

The findings of this manuscript are interesting and would be relevant for researchers interested in

understanding the task-dependent configuration of neural populations. I think it is an important addition to an increasing body of literature that focuses on cellular computations in the prefrontal cortex during adaptive behaviour. However, I have substantial comments that need addressing before the manuscript is considered for publication. My comments are below:

Major Comments:

1. The behavioural task used is reasonably simple, where two auditory tones are reversed in their association. The uncertain session is added during learning (only one session – i.e., 10% reward omission and 10% un-cued reward) where catch trials are incorporated. How many trials are these (20-30)? I am a little concerned that many of the key results are concluded based on a small number of trials. Can the authors clarify?

I am a little curious about why all animals completely stopping of licking after the reversal (Fig. 1f). Wouldn't some lick randomly in order to win a reward? Again, the drop in d' is caused by a decrease in 'miss' trials in only one session (uncertain session). Did the authors consider recording additional sessions? I am concerned about the T1-T3 comparison within one uncertain session (e.g., Fig. 2c-d). This will be significantly affected by animals' motivation, and thus, any comparison across T1 and T3 needs to be careful.

2. In Fig 4 and 1, only sensory stimulus-related responses are described here – the authors did not seem to separate outcome responses (No-go to FA and Miss). There was no separation between Go and CR, and they were always lumped together under responses to go cue. Would they see a difference if they break down go and No-go to specific trial types?

Could the decrease in responses be accounted for with a decrease in licking? If ACC neurons encode motor variables, the decrease in licking altogether (4c) could explain why neuronal responses were the same in all trials (go and No-go cues).

3. Fig 1 and 2: ACC neurons do not directly show increased responses to the Go cue in uncertain and reversal. Why do Hit responses in ACC after reversal go down? Do they completely lose selectivity? Reversal induces increased Hit responses in other prefrontal structures (e.g., OFC, see recent papers Banerjee et al. and Schoenbaum lab papers) – ACC doesn't seem to show this. Is ACC following this response or driving this? Please discuss.

4. Did the authors consider looking at A1 projecting ACC neurons (by retrograde genetic labelling)? Would a significant fraction of valence coding or dual function neurons fall into this category? Is this already known in the literature (compared to, say, V1 projecting ACC neurons – see works from Huda Lab (work with M. Sur), Adil Khan lab)? I think this would be really interesting and will strengthen the conclusions of this paper.

5. Outcome monitoring neurons were defined as those with mean responses in the RO trials higher than in Hit trials by at least 1.5 folds of S.D. – is that a significant and strong classification?

6. Difference of RO to hit across phases- Is the difference present already in the early phase or only in the late phase? Is it caused by a decrease in the hit or an increase in RO? Does exposure to RO decrease the value of Hit? Or would additional exposure to stable tasks lead to a devaluation of HIT outcomes as well?

7. "Line 189: Although the dual-function neurons only took a small portion here (78 out of 549 neurons), the decoding accuracy dropped significantly when they were removed from the population (Fig. 3h), indicating that they played an important role in computing outcomes and tracking experience history. The dual-function neuron might be a natural computation and storage element." - Does this really indicate history tracking? Does a high classification precision not only indicate a degree of immediate decoding (Fig. 3g higher value in the current trial if RO) but does not mean this is part of a "tracking" mechanism? Or as storing the information? Do they change over time, and the difference increases?

8. Line 227: Why are FA responses in the early phase increased, but No-go responses in the late phase decreased? In both, we see that early is higher than late, but this wording indicates different

mechanisms. Why cannot both be showing the same (e.g., increased activity immediately after Reversal? FA trials are a subset of No-go cues trials, after all). I am not sure if there is enough evidence to state "the No-go stimulus-evoked response was significantly reduced in the late period due to the cumulative devaluation of the previous reward-associated stimulus." Can the authors clarify?

9. Fig 5j,k why is t-1 opposite to t-2?

Minor Comments:

1. The manuscript is well written. However, I believe it will benefit from thorough proofreading. At times, the concepts are quite difficult to parse out and grasp and are presented in a rather convoluted way.

2. The configuration of recording ACC neurons was not mentioned clearly – I believe an imaging configuration in Fig. 1g would be helpful.

3. Line 142: Not sure if the authors can state - based on these results that motivation did not decrease. Please comment and edit.

4. Fig 3e: The $\Delta F/F$ changes around 5% - how are these values (strongly) significantly different?

5. Line 204: how is 1d relevant there?

6. Line 216/Fig 4e: immediate response to the stimulus itself seems similar, but there is a later decrease (around 0.25s post stimulation) – how do the authors explain this?

Reviewer #3 (Remarks to the Author):

General Remarks

This study approaches some important unresolved questions regarding the neural mechanisms underlying flexible decision making, focusing on an area of the cerebral cortex--the anterior cingulate--with established functional roles in action selection, performance monitoring, and the representation of task outcomes. Namely, the authors ask (1) if the ACC encodes changes in the contingencies between environmental cues and the outcomes they predict; (2) how information related to stimulus value and/or the corresponding error signals are represented in ACC neurons; and (3) how cognitive demand might dynamically affect the population recruitment patterns. To address these questions they employed in vivo cellular resolution calcium imaging during an auditory go/no-go behavioral paradigm with manipulated stimulus-reward contingencies (contingency degradation and reversal), as well as optogenetic suppression of ACC activity during these manipulations.

Throughout the manuscript, bold conclusions are drawn with insufficient support from the experimental evidence presented. Additionally, more care should be taken to note the specific measure or comparison used to support each substantive claim made in the text. In many cases, the figure panel cited to support an important claim does not directly represent the result described in the text. Beyond these fundamental weaknesses, the scientific rigor of this study would improve greatly if more efforts were taken to account for motor variables (especially anticipatory licking in response to cue onset), which may confound interpretation of the neural data.

Major Comments

1. Throughout the manuscript, an emphasis is placed on the neural responses to the two conditioned stimuli, and changes in neural activity during this period are interpreted to represent changes in the respective incentive values of the cues. However, the latent values associated with each cue are correlated with motor responses (lick bouts) that occur during the cue interval and could equally

account for the neural activity. A comparison of neural vs. behavioral response latencies relative to CS onset (Fig. S3) confirmed that neural responses occur earlier (median, 230 vs. 300 ms). However, this difference is overstated in the text (“...onset of go tone evoked response was much earlier than the licking onset”), and it does not rule out the possibility that the neural responses are at least partly motor-related (eg, preparatory activity is well-documented in neighboring areas of the frontal lobe such as M2/MOs). Moreover, the dF/F time series were temporally filtered at 5 Hz prior to analysis, which could affect the result. One solution would be to account for motor and cognitive factors simultaneously by incorporating them into a common statistical model (e.g. multiple linear regression) with the neural responses as the output variable. Another solution could be to amend the behavioral task to enforce a delay or trace period following the sound cue, during which time licking is penalized.

2. The authors claim to have found that

“a subpopulation of individual ACC neurons could reliably integrate the outcome information to the value representation of the stimulus in next-run trials.”

“The error-induced dynamic recruitment of such ACC neurons determined the impact of error signal on the iteration of value updating and the speed of decision adaptation, forming a non-linear feedback driven updating system to secure the appropriate decision switch.”

“Optogenetically suppressing ACC activity did not interfere the behavior performance with the established strategy, but significantly slowed down the feedback-driven decision switching”

These were the main conclusions of the study. However, as far as I can tell, no direct evidence was offered to support any of these claims. For (1) and (2), the claim is about changes between trials—so of course a trial-by-trial analysis would be required. Instead, several analyses were presented where the session is split in thirds, with changes in neural responses by the last third of the session (“T3”) presented as evidence for “next-run” updating. There could be many alternative explanations for changes like these over the course of a behavioral session, e.g. satiety or even drift in the neural representations. For (3), to demonstrate that learning was hampered, it would be necessary to compare learning rates in some way (number of trials-to-criterion, etc.), but no such comparison is given. Furthermore, Fig. 6h focuses on the FA rate to show some modest impairment in task performance—what about hit rate? From the example session, the most obvious difference between Stable and Reversal sessions is that hit rate approaches zero (regardless of opsin or control group). The measures presented in this figure could appear “cherry-picked” because 6b compares hit-rates, while 6e compares d' and hit-rate, and finally 6h compares only d' and the lick-responses to the CS+ (stable sessions) vs. the new CS- (Reversal). It would be better to pick one metric (d' , FA rate, hit rate, etc.) and use it for comparisons throughout—or better, to present all three for 6b,e,h.

3. A conceptual model for value updating is offered in Figure 1a, but no efforts were made to demonstrate a match between this model and the neurobehavioral data. A rigorous behavioral model could help clarify some of the most important issues in this study. For example, a reinforcement learning model (eg, Q-learning) could provide trial-by-trial estimates of the incentive values ascribed to each cue, as well as the associated prediction errors, to be used (in conjunction with motor variables, etc) in the statistical assessment of factors contributing to the neural activity.

Minor Comments

The writing was very difficult to understand at times.

The terms “value” and “valence” are used interchangeably throughout, but they do not mean the same thing. Value is graded and valence reflects positive or negative affective impact (or association with approach/avoidance) irrespective of magnitude.

Figure 4g: The distribution of consummatory lick-times relative to the reward has a distinctive multi-modal shape (with peaks at ~ 150 and 500 ms and a prominent trough at ~300 ms). Keeping in mind that the typical lick pattern of C57/Bl6 mice is very stereotyped with a frequency of ~7 Hz within each bout and a corresponding inter-lick interval of ~140 ms (for example, see Raymond et al., 2018), how would the authors explain this? Could there be a problem with the method of lick detection?

Raymond MA, Mast TG, Breza JM. An open-source lickometer and microstructure analysis program. *HardwareX*. 2018 Oct 1;4:e00035.

Zagha E, Erlich JC, Lee S, Lur G, O'Connor DH, Steinmetz NA, Stringer C, Yang H. The importance of

accounting for movement when relating neuronal activity to sensory and cognitive processes. *Journal of Neuroscience*. 2022 Feb 23;42(8):1375-82.

We thank the reviewers for their very constructive feedback and great suggestions, which have helped improve the manuscript. We have performed all of the suggested experiments and analyses and revised our manuscript accordingly. We have added a few new panels and Supplementary Figures as suggested by the reviewers.

One common question raised by both reviewers is that whether the neural response that we overserved in our experiment was caused by the motor viability of the licking behavior (major comment 2b from reviewer #2 and major comment 1 from reviewer #3). Specifically, the reviewers asked 1) whether the change of response is due to the change of licking behavior; 2) that it was not sufficient to rule out the correlation between the motor variable of licking and neural activity just based on the ~200 ms earlier onset of the neural response than the licking; 3) to account for motor and cognitive factors simultaneously by incorporating them into a common statistical model with the neural responses as the output variable. We addressed this issue by further analyzing the stimulus-evoked response in trials with early- and late-onset licking behavior in both the Uncertain and Reversal sessions, and used a general linear model (GLM) to estimate the contributions of licking and cognitive factors to the neural response. We included these analyses results in **Supplementary Fig. 3** and additional description in the Results session of the revised manuscript.

We analyzed the relationship between neuron activity and licking behavior in the Uncertain and Reversal sessions. The Hit trials in the Uncertain session were divided into two groups: early licking and late licking trials, corresponding to the trials in which anticipatory lick latency fell into the first and last quartile of the entire session (**Supplementary Fig. 3e**). The lick latency in the late licking trials was much longer than in the early licking trials (**Supplementary Fig. 3f**), but the onset of the go cue evoked response was similar in both types of trials (**Supplementary Fig. 3g**). Similarly, the number of anticipatory licks in the stimulus window was much lower in the late licking trials, but the mean amplitudes of tone evoked responses were similar in both groups (**Supplementary Fig. 3h-i**). We performed a similar analysis to the FA trials in the Reversal session (**Supplementary Fig. 3j-n**). The results were similar despite that the number of FA trials in the Reversal session was much smaller. We also used a general linear model (GLM) to estimate the contributions of task and behavioral conditions to the neural activity, including tone stimulus, cumulative trial history, previous outcome and licking rate (**Supplementary Fig. 3o**). Using this encoding model, we predicted neural activity by all variables or by excluding one of them, and

the relative contribution of each variable is determined by comparing how much the explained variance decreases when that variable is removed. The contribution of licking rate to the neural activity was only 0.02%, and was the lowest compared to the other three trial conditions (**Supplementary Fig. 3p**). Altogether, the modulation of tone evoked neural activity of ACC observed during the experiment was unlikely caused by the change in motor activity of licking behavior, as both the onset and amplitude of the neural activity were independent of licking, and the contribution of licking behavior to the neural activity was much smaller than the other cognitive factors.

The new paragraph in the results section states these results as follows (Line 98-101):

“The response during the stimulus window was not likely caused by licking upon the go cue, since the response was much earlier than the licking onset in the stimulus window, and neither the onset nor amplitude of neural response correlated with licking (Supplementary Fig. 3).”

We added a description of GLM in Methods of the revised manuscript (Line 605-617) as:

“To quantify the contribution of task variables to neural activity, we fitted a generalized linear model (GLM) for go trial during the Uncertain session⁵². In brief, we used four task variables (stimulus, cumulative history, previous outcome, and licking rate) to fit the per trial $\Delta F/F$ signal during stimulus window (0–1 s after the onset of the go cue). The stimulus variable was set to ‘1’ at all time points. The cumulative history variable ranges from 0 - 1, and represented the percentage of the total number of previously experienced unexpected trials (RO and UR). The previous outcome history variable was set to ‘1’ if the previous trial was unexpected outcome, including RO and UR trial or ‘0’ if the previous trial was normal outcome, respectively. The relative contribution of each task variable was quantified by calculating how much the explained variance decreased when each variable was removed from the model compared to using all of the variables (5-fold cross-validation was used in all cases).”

Supplementary Fig. 3 ACC activity in the stimulus window is not determined by licking. **a** The single-trial activity of two example neurons in Hit trials of the Stable session. Vertical black lines indicate the onset of the go cue and vertical white lines indicate the onset of the first lick. **b** Heat maps of the trial-averaged response of all recorded neurons

during Hit trials. Each row represents one neuron and they are aligned to the onset of the stimulus (left) and aligned to the first anticipatory lick (right). Colors bars on the left of the heat map indicate the neurons with activated (red), suppressed (blue), and non-response (gray) activity, respectively. **c** Population-averaged Ca^{2+} traces of three types of neurons aligned to the onset of the stimulus (left) and the first anticipatory lick (right). **d** The histogram of neural response and the first anticipatory lick latency relative to the onsets of stimulus ($n = 191$ trials from 5 mice, Wilcoxon rank-sum test). The neuron responses are locked to go cue onset rather than anticipatory licking. **a-d** data from the Hit trials in the Stable session. **e** Top: histogram showing the distribution of anticipatory licking onset time of an example mouse of Uncertain session. All Hit trials were divided into early (blue) and late (red) lick trials according to the upper and lower quartiles of the distribution. The blue, gray, and red triangles at the top represent the upper quartile, median, and lower quartile, respectively. Bottom: lick raster of the same mouse, sorted by onset times of the first lick. Blue shade: early licking trials, red shade: late licking trials, gray shade: other trials. **f** Anticipatory lick latency of early and late licking trials. (early licking: $n = 76$ trials from 6 mice, late licking: $n = 52$ trials from 6 mice. Wilcoxon rank-sum test). **g** The histogram of neural response latency of early and late licking trials ($n = 549$ neurons from 6 mice, Wilcoxon rank-sum test). **h** The anticipatory lick number in each early and late licking trial (Wilcoxon rank-sum test). **i** Left, average population responses to go stimulus of early (blue) and late licking trials (red). Black dots indicate the time segments when the responses are different ($P < 0.05$, Wilcoxon rank-sum test). Right: mean population response of different trials ($n = 549$ neurons from 6 mice, Wilcoxon sign-rank test). **j-n** Similar to **e-i**, but the analysis for the FA trials of Reversal session. **o** Schematic of the GLM model used to quantify the relationship between different variables and neuron response during go cue presentation in the Uncertain session. Data from the go trials in the Uncertain session. The model includes a total of four variables, where the stimulus variable is set to '1' at all points in time. The cumulative history variable represents the percentage of the total number of unexpected trials (RO and UR) that the animal has previously experienced, ranging from 0~1. The previous outcome history variable was '1' if it is the RO or UR trial, or '0' if it is the other trial type. The licking rate represents the frequency of licks at each point in time. Inset: predicted (dark blue) and actual (gray) $\Delta F/F$ signal for an example neuron ($R^2=0.47$). The neural activity was predicted with the GLM by all variables or by excluding one of them, and the relative contribution of each variable is determined by comparing how much the explained variance decreases when that variable is removed. **p** Left: relative contribution

of each variable to the explained neural activity variance (n = 549 neurons from 6 mice, Kruskal-Wallis test with post hoc Bonferroni multiple comparisons). Right: the distribution of the relative contribution of each variable. The different colored triangles at the top represent the median of the relative contribution of each variable for all neurons. Shading shows the standard error of the mean. In box plot: center line, median; box limits, upper and lower quartiles; notch limits, $(1.57 \times \text{interquartile range}) / \sqrt{n}$; whiskers, $1.5 \times \text{interquartile range}$. Outliers are not represented. See Table S1 for detailed statistics.

We have also identified 4 mistakes in the manuscript figures and corrected them in the revised manuscript.

- 1) In Fig. 4c (bottom), the licking probability in T1 was incorrectly calculated due to the typo in the analysis script. The panel and reported statistics have been updated. The associated figure legends were not affected and remain unchanged.
- 2) In Fig. 5c (top), the licking probability was incorrect due to the typo in the analysis script. The panel has been updated. The statistical analyses and the associated figure legends were not affected and remain unchanged.
- 3) In Supplementary Fig. 4g (left), the lick rate trace was shifted by ~ 0.2 s and an artifact was introduced at the beginning of the trace due to 3 Hz filtering of the raw lick rate trace. The panel has been updated. The statistical analyses and the associated figure legends were not affected and remain unchanged.
- 4) In Fig. 4g, the typo of the panel title was corrected in the revised manuscript.

We address all other reviewer comments below:

REVIEWER COMMENTS:

Reviewer #1 (Remarks to the Author):

Major points:

1: A major concern of this manuscript is a poor integration of the imaging data (Fig. 1-5) and the optogenetic manipulation study in Fig.6. While it is interesting that the optogenetic suppression of ACC neurons during cue period prevented the behavioral adjustment (Fig6e), the findings appear to be inconsistent to the imaging data showing reduced cue activity in uncertain condition (Fig. 2e, Fig3e). It looks to me that the findings of optogenetic manipulation do not support authors' working model. More thorough interpretation of the data and ideally bidirectional manipulation of cue period will help better understand the causal role of ACC neurons' activity changes in response to cues.

Thanks a lot for pointing out the confusing part. We should have explained it more clearly. The representation of the established stimulus-value relationship was relatively robust, and probably encoded in multiple brain areas for redundancy. Meanwhile, the iteration of such relationship under the conditions of subtle negative feedback, such as the Uncertain session, maybe more delicate and require much more sophisticated network computation. Perturbing the activity of ACC neurons during the cue period could disrupt the process of value renewal without abolishing the established value representation of the cue. This was demonstrated by the result of optogenetic manipulation of ACC neurons in the Stable session (see **Fig. 6b** in the manuscript), where neither the Hit rate nor the d' was affected by suppressing ACC activity in randomly selected 50% of the go trials.

We also performed a positive manipulation experiment as kindly suggested by the reviewer. We first tested the behavior response to optogenetic activation (ChR2) in the absence of a sound stimulus (see **Rebuttal Fig. 1**). To our surprise, spontaneous licking behavior was suppressed during optogenetic activation and reliably elicited when the light was off. The licking latency relative to light onset in ChR2 mice was 1.3374 ± 0.1848 s (mean \pm s.d.) with a light-on duration of 1 s. Meanwhile, optogenetic suppression of ACC neurons without sound stimulus did not affect the spontaneous licking behavior. Optogenetic activation of ACC during the go cue stimulus window also strongly suppressed anticipatory licking, and elicited strong rebound licking when the light was off, whereas suppressing ACC activity during the go cue did not affect

licking. Strong Chr2 mediated optogenetic activation of ACC excitatory neurons could hijack the brain and trigger the reflex-like licking behavior immediately after the light was off. It is likely an independent mechanism apart from the feedback-driven fine value iteration discussed here.

Rebuttal Fig. 1 Optogenetic activation of ACC neurons triggers licking response. a. Lick raster of mCherry (top), Chr2 (middle), and eNpHR mice (bottom). blue shade: optogenetic stimulation (488nm, 20 Hz, duration: 1 s); yellow shade: optogenetic stimulation (589nm, duration: 1 s). **b.** Trial-averaged licking frequency of mCherry, Chr2, and eNpHR mice. Light green shade: optogenetic stimulation. **c** Licking probability of pre-, during and post- optogenetic stimulation. The Chr2 group show higher licking frequency post optogenetic stimulation, illustrating that the photoactivation of ACC triggered rebound licking (mCherry: n = 6 mice; Chr2: n = 5 mice; eNpHR: n=4 mice. Two-way repeated measures ANOVA with post hoc Bonferroni multiple comparisons. Group: $F(2, 12) = 5.965$, $P = 0.0159$; time: $F(2, 24) = 6.590$, $P = 0.0052$; mCherry-post vs. Chr2-post, $***P = 4.3401e-05$; Chr2-post vs. eNpHR-post, $***P = 1.8656e-05$). **d-e** Optogenetic manipulation ACC activity during 3 kHz stimulus presentation in randomly selected 50% of go trials of the Stable session. **d** Raster plots of representative animals from the mCherry (left), Chr2 (middle), and eNpHR group (right) during light on and light off trials. Dark and light gray bars mark the stimulus and response window, respectively. **e** Lick rate during stimulus (left) and response windows (right). (mCherry: n = 6 mice; Chr2: n = 4 mice; eNpHR: n = 4 mice. Two-way repeated measures ANOVA with post hoc Bonferroni multiple comparisons. Stimulus window: group: $F(2, 11) = 7.713$, $P = 0.0081$; time: $F(1, 11) = 12.49$, $P = 0.0047$; mCherry-light on vs. Chr2-light on, $***P = 1.3517e-04$; Chr2-light on vs. eNpHR-light on, $***P = 2.3485e-04$). Error bars indicate the standard error of the mean.

2: Related to the above point, it is unclear why authors chose to manipulate ACC neurons during auditory cue phase, but not during outcome phase. As imaging data suggests the importance of outcome feedback (Fig.3a-c, Fig4g-i). I think it is essential

to examine the impact of outcome feedback period in the optogenetic manipulation study to support the authors' main conclusion.

We agreed that the outcome comparison was very critical for the feedback-guided value updating. We gladly performed the optogenetic manipulation during the response window suggested by the reviewer and the results were included in a new **Supplementary Fig. 13** in the manuscript. The ACC neuron activity was suppressed during the response window in the unexpected trials, including both RO and UR trials in the Uncertain session and FA trials in the Reversal session (**Supplementary Fig. 13 a, c**). The switching of performance strategy was delayed in both the Uncertain and Reversal sessions (see **Supplementary Fig. 13 b, d**), similar to the optogenetic suppression during the stimulus window.

Supplementary Fig. 13. Optogenetic suppression of ACC while experiencing the unexpected outcome impairs decision switching. **a**. Scheme of optogenetic suppression of the ACC neurons in the Uncertain session. 589 nm laser with a duration of 2 s is delivered after the reward omission (RO trial) or un-cued reward (UR trial) was triggered. **b**. Suppressing ACC activity when the animal experiencing the unexpected outcomes blocked the decision switch in the Uncertain session. Behavior performance for d' (left) and 3 kHz licking probability (Hit rate, right) of eNpHR suppressing and mCherry control group. (mCherry: $n = 4$ mice; eNpHR: $n = 4$ mice. Two-way repeated measures ANOVA, within-group comparisons: paired t-test with post hoc Bonferroni multiple comparisons; between-group comparisons: unpaired t-test with post hoc Bonferroni multiple comparisons). **c-d** Suppressing ACC activity when the animal experiencing the unexpected punishment slowed down the decision switch in the Reversal session. Similar to **a-b**, except optogenetic suppression was delivered after the air-puff was triggered (mCherry: $n = 4$ mice; eNpHR: $n = 4$ mice. Two-way repeated measures ANOVA, within-group comparisons: paired t-test with post hoc Bonferroni multiple comparisons; between-group comparisons: unpaired t-test with post hoc Bonferroni multiple comparisons). $*P < 0.05$, $**P < 10^{-2}$, $***P < 10^{-3}$. Error bars indicate the standard error of the mean. See Table S1 for detailed statistics.

Other points:

3: Please describe the quantitative criteria to define “outcome monitoring neurons” and “valence updating neurons” in Fig.3 and 4.

We defined “outcome monitoring neurons” and “value updating neurons” by comparing the response of each neuron in different trials within the same session. More specifically, (1) in Figure 3 (during the Uncertain session), outcome monitoring neurons were defined as:

$$\overline{R_{RO}} > \overline{R_{Hit}} + 1.5 * SD_{Hit}$$

where $\overline{R_{RO}}$ is the mean activity during the response window in RO trials, $\overline{R_{Hit}}$ is the mean activity during the response window in Hit trials, and SD_{Hit} is the standard deviation of activity during the response window in all Hit trials.

Value updating neurons were defined as:

$$\overline{R_{go_early}} > \overline{R_{go_late}} + 1.5 * SD_{go_late}$$

where $\overline{R_{go_early}}$ is the mean activity during the stimulus window in the go trials in the early period, $\overline{R_{go_late}}$ is the mean activity during the stimulus window in the go trials in the late period, and SD_{go_late} is the standard deviation of the activity during the stimulus window in the late period.

It is also explained in the manuscript (Line 548-554) as

“Outcome monitoring neurons were defined as those neurons whose mean response in the RO trials was higher than those in Hit trials by at least 1.5 folds of s.d.

...

Value updating neurons were defined as those neurons whose go stimulus evoked mean response in the early period was higher than that in the late period by at least 1.5 folds of s.d.”

(2) in Figure 4 (during the Reversal session), outcome monitoring neurons were defined as:

$$\overline{R_{FA_early}} > \overline{R_{FA_late}} + 1.5 * SD_{FA_late}$$

where $\overline{R_{FA_early}}$ is the mean activity during the response window in FA trials in the early period, $\overline{R_{FA_late}}$ is the mean activity during the response window in FA trials in the late period, and SD_{FA_late} is the standard deviation of the activity during the response window in the late period.

Value updating neurons were defined as:

$$\overline{R_{no-go_early}} > \overline{R_{no-go_late}} + 1.5 * SD_{no-go_late}$$

whereas, $\overline{R_{\text{no-go_early}}}$ is the tone-evoked mean activity during the stimulus window in no-go trials in the early period, $\overline{R_{\text{no-go_late}}}$ is the tone-evoked mean activity during the stimulus window in no-go trials in the late period, and $SD_{\text{no-go_late}}$ is the standard deviation of the activity during the stimulus window the late period.

It is also explained in the manuscript (Line 560-565) as

“Neurons with a mean air-puff evoked response that was higher in the early period than that in the late period by at least 1.5 folds of s.d. were defined as outcome monitoring neurons. Neurons with a mean no-go cue evoked response during 1-s window after stimulus onset) in the early period that was higher than that in the late period by at least 1.5 folds of s.d. were defined as value updating neurons.”

Rebuttal Fig. 2 Analyses results of alternative criteria for outcome monitoring and value updating neurons ($\Delta R_{\text{peak}} > 4$ s.d.). a-c Identified outcome monitoring neuron and value updating neuron in Uncertain session. **a** Left: trial-averaged activity of the outcome monitoring neurons in reward omission (RO) and Hit trials during the Uncertain session. White vertical lines denote reward delivery time. (n = 273 neurons from 6 mice, each row

represents one neuron, and they were sorted by response profiles). Right: mean population responses of outcome monitoring neurons during 1-s window after reward omission or delivery during RO and Hit trials (Wilcoxon signed-rank test, $***P = 1.9209e-17$). **b** Left: trial-averaged go cue-evoked response of value updating neurons in the early (T1 + T2, left) and late (T3, right) periods. White vertical lines denote the stimulus onset. (n = 88 neurons from 6 mice, each row represents one neuron, and they were sorted by response profiles). Right: mean population responses of value updating neurons during 1-s window after cue onset in early and late periods (Wilcoxon signed-rank test, $***P = 4.0067e-10$). **c** Venn diagram of the number of neurons identified to perform outcome monitoring (light orange) and value updating (light blue) functions. **d-f** Similar to **a-c**, analysis for the Reversal session. **d** Left: trial-averaged activity of outcome monitoring neurons in response to the air-puff in the early (T1, left) and late (T2 + T3, right) periods. White vertical lines denote air-puff delivery time. (n = 208 neurons from 5 mice, each row represents one neuron, and they were sorted by response profiles). Right: mean population responses of outcome monitoring neurons during 1-s window after air-puff delivery in the early and late periods (Wilcoxon signed-rank test, $***P = 1.9457e-13$). **e** Left: trial-averaged activity of value updating neurons in response to the no-go cue in the early and late periods. White vertical lines denote the stimulus onset. (n = 348 neurons from 5 mice, each row represents one neuron, and they were sorted by response profiles). Right: mean population responses of value updating neurons to the no-go cue in the early and late periods (Wilcoxon signed-rank test, $***P = 7.0731e-36$). **f** Venn diagram of the number of neurons identified to perform outcome monitoring (light orange) and value updating (light blue) functions. **g** Venn diagram of the number of outcome monitoring neurons (left) and value updating neurons (right), identified in the Uncertain (light gray) and Reversal sessions (black). Note that 56 neurons recorded in the Uncertain session were not able to be identified in the Reversal session. In box plot: center line, median; box limits, upper and lower quartiles; notch limits, $(1.57 \times \text{interquartile range}) / \sqrt{n}$; whiskers, $1.5 \times \text{interquartile range}$. Outliers are not represented.

We also tried other criteria to quantify outcome monitoring neurons and value updating neurons. For example, we tried to quantify the peak response of each cell (see **Rebuttal Fig. 2**), and define outcome monitoring neurons in the Uncertain session as:

$$R_{RO} > R_{Hit} + 4 * SD_{Hit}$$

outcome monitoring neurons in the Reversal session as:

$$R_{FA_early} > R_{FA_late} + 4 * SD_{FA_late}$$

value updating neurons in the Uncertain session as:

$$R_{go_early} > R_{go_late} + 4 * SD_{go_late}$$

and value updating neurons in the Reversal session as:

$$R_{no-go_early} > R_{no-go_late} + 4 * SD_{no-go_late}$$

We also tried to use more restrictive criteria with the mean response + 2.5-fold standard deviation (see **Rebuttal Fig. 3**), and define outcome monitoring neurons in the Uncertain session as:

$$\overline{R_{RO}} > \overline{R_{Hit}} + 2.5 * SD_{Hit}$$

outcome monitoring neurons in the Reversal session as:

$$\overline{R_{FA_early}} > \overline{R_{FA_late}} + 2.5 * SD_{FA_late}$$

value updating neurons in the Uncertain session as:

$$\overline{R_{go_early}} > \overline{R_{go_late}} + 2.5 * SD_{go_late}$$

value updating neurons in the Reversal session as:

$$\overline{R_{no-go_early}} > \overline{R_{no-go_late}} + 2.5 * SD_{no-go_late}$$

Results yielded from different quantification criteria of outcome monitoring neurons and value updating neurons were similar. More dual-function neurons were recruited in the Reversal session.

Rebuttal Fig. 3 Analyses results of alternative criteria for outcome monitoring and value updating neurons ($\Delta R_{mean} > 2.5$ s.d.). a-c Identified outcome monitoring neuron and value updating neuron in Uncertain session. a Left: trial-averaged activity of the outcome monitoring neurons in reward omission (RO) and Hit trials during the Uncertain

session. White vertical lines denote reward delivery time ($n = 169$ neurons from 6 mice, each row represents one neuron, and they were sorted by response profiles). Right: mean population responses of outcome monitoring neurons during 1-s window after reward omission or delivery during RO and Hit trials (Wilcoxon signed-rank test, $***P = 5.6454e-17$) **b** Left: trial-averaged go cue-evoked response of value updating neurons in the early (T1 +T2, left) and late (T3, right) periods. White vertical lines denote the stimulus onset. ($n = 28$ neurons from 6 mice, each row represents one neuron, and they were sorted by response profiles). Right: mean population responses of value updating neurons during 1-s window after cue onset in early and late periods (Wilcoxon signed-rank test, $***P = 1.1083e-05$). **c** Venn diagram of the number of neurons identified to perform outcome monitoring (light orange) and value updating (light blue) functions. **d-f** Similar to **a-c**, analysis for the Reversal session. **d** Left: trial-averaged activity of outcome monitoring neurons in response to the air-puff in the early (T1, left) and late (T2 + T3, right) periods. White vertical lines denote air-puff delivery time. ($n = 154$ neurons from 5 mice, each row represents one neuron, and they were sorted by response profiles). Right: mean population responses of outcome monitoring neurons during 1-s window after air-puff delivery in the early and late periods (Wilcoxon signed-rank test, $***P = 2.2721e-09$). **e** Left: trial-averaged activity of value updating neurons in response to the no-go cue in the early and late periods. White vertical lines denote the stimulus onset. ($n = 158$ neurons from 5 mice, each row represents one neuron, and they were sorted by response profiles). Right: mean population responses of value updating neurons to the no-go cue in the early and late periods (Wilcoxon signed-rank test, $***P = 9.8058e-26$). **f** Venn diagram of the number of neurons identified to perform outcome monitoring (light orange) and value updating (light blue) functions. **g** Venn diagram of the number of outcome monitoring neurons (left) and value updating neurons (right), identified in the Uncertain (light gray) and Reversal sessions (black). Note that 56 neurons recorded in the Uncertain session were not able to be identified in the Reversal session. In box plot: center line, median; box limits, upper and lower quartiles; notch limits, $(1.57 \times \text{interquartile range}) / \text{sqrt}(n)$; whiskers, $1.5 \times \text{interquartile range}$. Outliers are not represented.

4: Fig.2e: It would be informative if authors describe how the post unexpected outcome activity changes as a function of the time. It would be helpful if they can show the breakdown with T1, T2, T3 as in Fig2d.

Thanks for the suggestion. We analyzed the go cue evoked activity in the post normal and post unexpected outcome trials through T1-T3 in the Uncertain session and added the analysis results in the **Supplementary Fig. 4e-g** in the revised manuscript. We found that the response gradually decreased with time in both the post normal and post unexpected trials (**Supplementary Fig. 4f**). The Hit rate of the post unexpected and normal outcome trials decreased only slightly with time (no significant difference,

Supplementary Fig. 4g). Notably, the go cue evoked response in the post-unexpected outcome trials was significantly lower than that in the post normal trials in T1 phase (**Supplementary Fig. 4f, \$**), but the Hit rate of both types of trials was very similar in T1 (**Supplementary Fig. 4g**). The change of neural activity in the post unexpected outcome activity along T1, T2, T3 was similar to the trend of all trials in the Uncertain session (**Fig. 2** in the manuscript), suggesting that the changes of stimulus-evoked response in the post unexpected outcome and post normal outcome trials were both gradual in the Uncertain session.

Supplementary Fig. 4e-g Trial-averaged population responses to go stimulus in trials following unexpected outcomes (red, including RO and UR trials) and following other normal outcomes (green) in the T1-T3 phases of the Uncertain session. Black dots indicate the time segments when the activity is different from each other ($P < 0.05$, Wilcoxon rank-sum test). **f** Mean response to go stimulus following different outcome in the T1-T3 phases of the Uncertain session ($n = 549$ neurons from 6 mice, Friedman test with post hoc Bonferroni multiple comparisons). The symbol * denotes significant differences between the trials following the same outcomes in different phases, and the symbol \$ denotes significant differences between the trials following different outcomes in the same phase). **g** The Hit rate of trials following unexpected outcomes (red) and other normal outcomes (green) across the T1-T3 phase. ($n = 6$ mice, two-way repeated measures ANOVA with post hoc Bonferroni multiple comparisons).

5: Similarly, Fig 3 bc can be break out to T1-3 to see if there is any time dependent changes.

We analyzed the activity during the response window in T1, T2, and T3 phases and added the analysis results to the **Supplementary Fig. 7** in the revised manuscript. The

RO-evoked response was higher than the expected reward in both the whole population, as well as in the outcome monitoring neurons throughout the whole session (except for the T1 phase for the whole population). Interestingly, the outcome evoked response was significantly higher in the T3 phase for both RO and Hit trials, indicating that the experience of unexpected events might lead to an upregulation of the representation of all outcomes, and therefore enhanced the error detection function in the later stage of the Uncertain session.

Supplementary Fig. 7 Change in response of Hit and RO with time. **a** Mean response of all neurons during 1-s window after reward omission or delivery during RO and Hit trials in the T1-T3 phases of the Uncertain session (n = 549 neurons from 6 mice, Friedman test with post hoc Bonferroni multiple comparisons). **b** Mean responses of outcome monitoring neurons during 1-s window after reward omission or delivery during RO and Hit trials in the T1-T3 phases of the Uncertain session (n = 260 neurons from 6 mice, Friedman test with post hoc Bonferroni multiple comparisons). * $P < 0.05$, *** $P < 10^{-3}$, \$\$\$ $P < 10^{-3}$. In box plot: center line, median; box limits, upper and lower quartiles; notch limits, $(1.57 \times \text{interquartile range}) / \sqrt{n}$; whiskers, $1.5 \times \text{interquartile range}$. Outliers are not represented. See Table S1 for detailed statistics.

6: It is helpful if authors document in which cortical layers the imaged neurons were located.

We recorded mainly L2/3 neurons in the ACC. The recording region was 300 μm beneath the pial. We added a scheme of recording setting in **Fig. 1d** and added a description of recordings in Methods of the revised manuscript (Line 483-485) as

“Imaging region of $\sim 250 \times 250 \mu\text{m}^2$ was located lateral to the superior sagittal sinus at a depth of 250 - 400 μm below the pial, corresponding to the posterior

dorsal ACC.”

Fig. 1 d Scheme (left) of imaging L2/3 ACC neurons through a cranial window (the grey horizontal bar) and a coronal section (right) of ACC neurons expressing GCaMP6f in the recorded mouse (post-mortem). The red dotted lines indicate L2/3 and L5 of the cortex, and the black horizontal line represents the plane of imaging. M2, the secondary motor cortex.

Reviewer #2 (Remarks to the Author):

Major Comments:

1. The behavioural task used is reasonably simple, where two auditory tones are reversed in their association. The uncertain session is added during learning (only one session – i.e., 10% reward omission and 10% un-cued reward) where catch trials are incorporated. How many trials are these (20-30)? I am a little concerned that many of the key results are concluded based on a small number of trials. Can the authors clarify?

I am a little curious about why all animals completely stopping of licking after the reversal (Fig. 1f). Wouldn't some lick randomly in order to win a reward? Again, the drop in d' is caused by a decrease in 'miss' trials in only one session (uncertain session). Did the authors consider recording additional sessions? I am concerned about the T1-T3 comparison within one uncertain session (e.g., Fig. 2c-d). This will be significantly affected by animals' motivation, and thus, any comparison across T1 and T3 needs to be careful.

Yes, we introduced only a small number of RO and UR trials in the Uncertain session, about 10 trials each. We kept the number of RO and UR trials small, because we wanted to test the response of the animals to small perturbances of certainty under the condition that they were initially very certain about the rule. Although the number of unexpected trials was low, the response of RO trials was consistent across T1, T2, and T3 phases during the Uncertain session.

We did not extend the Uncertain session to collect more trials, because we wanted to avoid the change in motivation to perform the task in the late trials. The licking behavior reflects the motivation of the animal to perform the task ¹. In our experiment, the animal's motivation was stable throughout T1-T3 in the Uncertain session. The number of anticipatory lick (during the stimulus window) and the consumption lick (after the valve opening in the response window) was unchanged across T1-T3 (**Supplementary Fig. 5** in the revised manuscript). The latency of the first lick was also unchanged across T1-T3. We have modified the description in the revised manuscript (Line 142-146) as:

“The experience of RO trials led to an adjustment in the decision strategy, but not a decrease in motivation to perform the task, as the lick latency, the number of anticipatory and the consumption lick rate were stable throughout T1-T3 phases in the Uncertain session and as high as in the Stable session

(Supplementary Fig. 5).”

Supplementary Fig. 5 Consistent performance of anticipatory and consumption licking during the Stable and Uncertain sessions. **a.** The anticipatory licking rate of the Hit trials in the Stable session and T1-T3 phases of the Uncertain session. **b** The number of anticipatory licks (left) and lick latency (right) in each phase (n = 81~191 trials from 6 mice. Kruskal-Wallis test with post hoc Bonferroni multiple comparisons). **c** The consumption licking rate relative to the onset of reward delivery in the Stable session and T1-T3 phases of the Uncertain session (left), and the number of total consumption licks in each phase (right, n = 71~191 trials from 6 mice. Kruskal-Wallis test with post hoc Bonferroni multiple comparisons). Shading shows the standard error of the mean. In box plot: center line, median; box limits, upper and lower quartiles; notch limits, $(1.57 \times \text{interquartile range}) / \sqrt{n}$; whiskers, $1.5 \times \text{interquartile range}$. Outliers are not represented. See Table S1 for detailed statistics.

Each animal was tested only once for the Uncertain session, because we did not want the animals to learn the possibility of an unexpected outcome as a secondary rule. We tested another group of animals with the paired Stable-Uncertain session for 7 days. The decrease in performance caused by the unexpected outcome recovered with more Uncertain training sessions, and the performance in the Uncertain session became similar to that in the Stable session (**Rebuttal Fig. 4**). The overall decrease of performance discrimination was only observed on the first day of the Uncertain training (**Rebuttal Fig. 4a**), and the time dependent decrease of performance discrimination and Hit rate were only observed on the first day too (**Rebuttal Fig. 4b-d**). It suggested that the animals were able to learn and adapt to the rule of unexpected reward omission and unexpected reward.

Rebuttal Fig. 4 The animals get habituated to the unexpected outcomes after repetitive training of the Uncertain sessions. **a** Performance discrimination (d') in repeated training of the paired Stable-Uncertain sessions. ($n = 6$ mice, paired t-test, Day1: $*P = 0.0131$). After the mice had reached a stable high performance ($d' \geq 1.5$ for at least three days), they were tested for the Stable (50%/50% randomized go/no-go task; 80 trials/session), and Uncertain session (40% of go trials, 40% of no-go trials, 10% of reward omission trials, and 10% of un-cued reward trials; 80 trials/session) daily for 7 days, with the trials were arranged randomly in each session. **b** Behavior performance (d') in the Stable session and T1-T3 phases (corresponding to the first, middle, and last 33% of trials, respectively) of the Uncertain session. ($n = 6$ mice, one-way repeated measures ANOVA with post hoc Dunnett's multiple comparisons, comparing each Uncertain phase with Stable session on the same day. Day1, Stable vs. T3: $**P = 0.0063$). **c-d** Similar to **a-b**, except performance measured as Hit rate (Day1, Stable vs. T3: $*P = 0.0428$). Error bars represent the standard error of the mean.

Most animals completely stopped licking in the late phase of the Reversal session, because they received air puff punishment for FA trials at the beginning of the Reversal session, and they became very conservative after a few consecutive punishments. Therefore, most animals tended to stop licking in the second half of the Reversal session. If the air puff punishment for FA was removed, the animals would lick much more often to take random chances to win the reward.

2a. In Fig 4 and 1, only sensory stimulus-related responses are described here – the authors did not seem to separate outcome responses (No-go to FA and Miss). There was no separation between Go and CR, and they were always lumped together under responses to go cue. Would they see a difference if they break down go and No-go to specific trial types?

In the manuscript, we pooled the no-go tone evoked response of the CR and FA trials together, because the 3 kHz tone evoked response in the CR and FA trials in the Reversal session had a similar shift trend. They both gradually decreased over time (see **Rebuttal Fig. 5**). The FA trials in the T2 and T3 phases were pooled together here, because some animals stopped licking completely in the T3 phase.

Rebuttal Fig. 5 Rapid value updates occur in both CR and FA trials in the Reversal session. **a.** Mean responses to 3 kHz tone in CR trials across the T1-T3 phases in the Reversal session ($n = 493$ neurons from 5 mice, Friedman test with post hoc Bonferroni multiple comparisons. T1 vs. T2: $***P = 5.3386e-06$; T1 vs. T3: $***P = 7.4430e-07$). **b** Mean responses to 3 kHz tone in FA trials in the T1 and T2+T3 phases in the Reversal session ($n = 493$ neurons from 5 mice, Wilcoxon signed-rank test $**P = 0.0080$). In box plot: center line, median; box limits, upper and lower quartiles; notch limits, $(1.57 \times \text{interquartile range}) / \sqrt{n}$; whiskers, $1.5 \times \text{interquartile range}$. Outliers are not represented.

We pooled Hit and Miss trials together as go trials, as well as for the no-go trials, because the animal made very few Miss or FA choices in the Stable and Re-stable session. We felt too biased to quantify the response of one type of choice based on only 1 or 2 trials. Here we analyzed the Hit, Miss, CR and FA trials separately and compared them to the lumped go and no-go cue evoked response (see **Rebuttal Fig. 6**). The animal that only exhibited 2 or less than 2 trials for the same choice in one session was excluded from the analysis for that particular trial type in that session. These trials were also removed from the population activity heat maps in **Supplementary Fig. 2a**. The tone evoked responses were similar in the Hit and lumped go trials in the Stable, Uncertain, and Re-stable sessions, and so were the responses in the CR trials and lumped no-go trials. In the Reversal session, the response in the Miss trials was similar to that in the lumped go trials, and so were those in the CR and the lumped no-go trials.

Rebuttal Fig. 6 Comparison of stimulus evoked responses between trials with different decisions. Neuronal responses to go and no-go cues in different decisions (Hit, Miss, CR, FA) in four sessions. For each type of decision, only the animals performed > 2 times were included for analysis (n = 493 neurons from 5 mice. Kruskal-Wallis test with post hoc Bonferroni multiple comparisons for each session. Uncertain go cue: total vs. Miss, $**P=0.0021$; Hit vs. Miss, $***P=1.1587e-04$; Reversal no-go cue: total vs. FA, $***P = 7.2278e-10$; CR vs. FA, $***P = 1.2487e-15$; Re-stable no-go cue: total vs. FA, $*P = 0.0251$; CR vs. FA, $*P = 0.0351$). In box plot: center line, median; box limits, upper and lower quartiles; notch limits, $(1.57 \times \text{interquartile range}) / \text{sqrt}(n)$; whiskers, $1.5 \times \text{interquartile range}$. Outliers are not represented.

2b. Could the decrease in responses be accounted for with a decrease in licking? If ACC neurons encode motor variables, the decrease in licking altogether (4c) could explain why neuronal responses were the same in all trials (go and No-go cues).

We appreciated that the reviewer pointed out the critical point that licking behavior could possibly cause the change in the cue-evoked response. We analyzed the relationship between neuron activity and licking behavior in the Uncertain and Reversal sessions in the first part of the rebuttal (addressed together with the major comment 1 of reviewer #3) and added the analysis results in the **Supplementary Fig. 3** in the revised manuscript. We found that the cue evoked response in ACC neurons was not closely correlated with the number or onset of licks during the stimulus window and the contribution of licking to the stimulus evoked response was much smaller than other cognitive factors.

3. Fig 1 and 2: ACC neurons do not directly show increased responses to the Go cue in uncertain and reversal. Why do Hit responses in ACC after reversal go down? Do they completely lose selectivity? Reversal induces increased Hit responses in other prefrontal structures (e.g., OFC, see recent papers Banerjee et al. and Schoenbaum lab papers) – ACC doesn't seem to show this. Is ACC following this response or driving this? Please discuss.

The Reversal session we recorded here was the first session of reverse learning. The new go cue (12 kHz tone) was reliably associated with punishment in the previous session, and the animal rarely licked upon the new go cue in the Reversal session. It

took much longer reverse learning for the animal to re-establish the association of the new go cue with reward. The animal received consecutive punishment when licking upon the new no-go cue (3 kHz tone) at the beginning of the Reversal session and quickly stopped licking and reduced the response to the new no-go cue (**Rebuttal Fig. 7a-b**). Choice discrimination (d') became close to zero in the T3 phase and the selectivity of the tone evoked response was gradually decreased and became non-selective in the T3 phase too (**Rebuttal Fig. 7c**). Note that although the number of go cue selected neurons increased in T2, perhaps because many neurons decreased their response to the new no-go cue due to repeated FA punishments, the absolute values of selectivity index of these neurons were relatively low and the responses elicited by go and no-go cues were both low (**Rebuttal Fig. 7b**).

Rebuttal Fig. 7 Significant decrease of selectivity of both behavior decision and neuronal response across T1-T3 in the Reversal session. **a** Left: performance discrimination (d') across the T1-T3 phases of Reversal session (n = 5 mice, one-way repeated measures ANOVA with post hoc Bonferroni multiple comparisons. F (1.955, 7.820) = 8.565, $P = 0.0109$. T1 vs. T3: $*P = 0.0326$. Right: performance of Hit and FA rate in T1-T3 phases. **b** Mean responses to go and no-go cues across the T1-T3 phases of the Reversal session (n = 493 neurons from 5 mice, Friedman test with post hoc Bonferroni multiple comparisons. no-go cue: T1 vs. T2: $***P = 2.9867e-16$; T1 vs. T3: $***P = 1.1685e-19$; go vs. no-go: T1, $***P = 8.4422e-07$; T3, $*P = 0.0144$). **c** Stimulus selectivity index (SI) of individual neurons in the T1-T3 phase of Reversal sessions (n = 493 cells from 5 mice, one-sample Wilcoxon signed-rank test, compared with 0: T1, $***P = 1.180e-15$; T2, $***P = 3.726e-10$). Color dots indices neurons with significant SI in each phase ($P < 0.05$, permutation test). Error bars indicate the standard error of the mean. In box plot: center line, median; box limits, upper and lower quartiles; notch limits, $(1.57 \times \text{interquartile range}) / \sqrt{n}$; whiskers, $1.5 \times \text{interquartile range}$. Outliers are not represented.

We added discussion about it in the manuscript (Line 329-336) as:

“Several brain areas, including ACC^{1,9,23}, the medial prefrontal cortex²⁴⁻²⁷ and striatum²⁸⁻³⁰, have been reported able to detect errors and encode value information. For example, the orbitofrontal cortex encoded reward and

prediction related information in goal-directed behaviors and the response of OFC neurons increased after reverse learning^{31,32}. Similar upregulated response and selectivity was also observed in ACC (Supplementary Fig. 2b and 9f), indicating that the ACC and other frontal cortical areas including OFC, forming a larger network providing necessary information to the decision-making process.”

4. Did the authors consider looking at A1 projecting ACC neurons (by retrograde genetic labelling)? Would a significant fraction of valence coding or dual function neurons fall into this category? Is this already known in the literature (compared to, say, V1 projecting ACC neurons – see works from Huda Lab (work with M. Sur), Adil Khan lab)? I think this would be really interesting and will strengthen the conclusions of this paper.

We injected RetroAAV-mCherry to A1 to label the A1-projecting neurons. The number of labeled ACC-A1 neurons was very low (**Rebuttal Fig. 8a-b**). The A1-projecting ACC neurons were mainly located in the anterior dorsal ACC rather than the posterior dorsal ACC². It was hard to conclude how information was encoded in so few neurons. We also retrogradely labelled ACC, the number labeled ACC-projecting A1 neurons was also very low (**Rebuttal Fig. 8d-e**), whereas there were more ACC-projecting neurons in the visual areas.

Rebuttal Fig. 8 Sparsely connected between ACC and Aud. **a** Injection of a retrograde

vector expressing mCherry into the auditory cortex (Aud). **b** Top left: representative image of back-labeled neurons in the ACC. Right: high magnification image. White arrows denote mCherry-expressing cells. **c** Similar to **b**, but showing back-labeled neurons in the medial geniculate nucleus (MG). **d** RetroAAV injection into ACC for retrograde tracing. **e** Representative image shows mCherry-expressing cells selectively observed in the visual cortex (VIS) more than the auditory cortex (Aud).

In addition, we recorded the response of A1 neurons during the task (see **Rebuttal Fig. 9**). The licking probability decreased in both the Uncertain and Reversal sessions, compared to the Stable session (**Rebuttal Fig. 9c, e**). The stimulus evoked response in A1 to both the go and no-go cue became much stronger in the Uncertain and Reversal sessions (**Rebuttal Fig. 9d, f**). The change in A1 activity was very different from that of ACC neurons, whose response was downregulated in the Uncertain and Reversal sessions. The activity of A1 neurons was boosted, perhaps by salient signal that was upregulated by experiencing unexpected outcomes, such as RO and air-puff in the Uncertain and Reversal sessions.

Rebuttal Fig. 9 Tone-evoked activity in auditory cortex is modulated by sensory salience. **a** schematic of *in vivo* two-photon imaging of the auditory cortex. Top right: schematic of the injection site. **b** Schemes of go/no-go tasks with modified stimulus-reward contingencies. **c** Behavioral performance in the Stable session and T1-T3 phases (corresponding to the first, middle and last 33% of trials, respectively) of the Uncertain session. ($n = 4$ mice, one-way repeated measures ANOVA with post hoc Tukey's multiple

comparisons. Hit rate, Stable vs. T3: $*P = 0.0163$). **d** Mean response to go (left, 3 kHz tone) and no-go stimulus (right, 12 kHz tone) in the Stable session and T1-T3 phases in the Uncertain session ($n = 1014$ neurons from 4 mice, Friedman test with post hoc Bonferroni multiple comparisons. $**P < 10^{-2}$, $***P < 10^{-3}$). **e** Licking probability in the Stable session and T1- T3 phases (corresponding to the first 15%, middle 35% and last 50% of trials, respectively) of the Reversal session ($n = 4$ mice, one-way repeated measures ANOVA with post hoc Tukey's multiple comparisons. 3 kHz licking, Stable vs. T2: $*P = 0.0103$; Stable vs. T3: $**P = 0.0026$). **f** Mean population response to 3 kHz (left) and 12 kHz stimulus (right) in the Stable session and T1-T3 in the Reversal session ($n = 1014$ neurons from 4 mice, Friedman test with post hoc Bonferroni multiple comparisons. $**P < 10^{-2}$, $***P < 10^{-3}$). Error bars represent the standard error of the mean. In box plot: center line, median; box limits, upper and lower quartiles; notch limits, $(1.57 \times \text{interquartile range}) / \sqrt{n}$; whiskers, $1.5 \times \text{interquartile range}$. Outliers are not represented.

We added discussion about the top-down regulation in the Discussions part in the manuscript (Line 391-397) as:

“The ACC collects and converts the sensory and feedback information to update value information, provides adjustment to optimize the decision-making process⁴¹. The updated value information is also very important top-down regulatory signals, which was sent back directly or indirectly to the sensory and motor cortices^{13,39,42} to guide the behavior-relevant sensory processing and memory formation. Such as in the primary visual cortex, the perceptual learning enhanced visual response and selectivity might be mediated by the precise top-down modulation from ACC⁴²”

5. Outcome monitoring neurons were defined as those with mean responses in the RO trials higher than in Hit trials by at least 1.5 folds of S.D. – is that a significant and strong classification?

We also tried other criteria to quantify outcome monitoring and value updating neurons. These yielded similar results. For example, we have tried to define outcome monitoring neurons as those with: (1) the peak responses in the RO trials higher than in Hit trials by at least 4 folds of s.d., and similar to the value updating neurons (see **Rebuttal Fig. 2**); or (2) the mean responses in the RO trials higher than in Hit trials by at least 2.5 folds of s.d., and similar to the value updating neurons (See **Rebuttal Fig. 3**)

Uncertain session

Reversal session

Rebuttal Fig. 2 Analyses results of alternative criteria for outcome monitoring and value updating neurons ($\Delta R_{\text{peak}} > 4$ s.d.).

a-c Identified outcome monitoring neuron and value updating neuron in Uncertain session. **a** Left: trial-averaged activity of the outcome monitoring neurons in reward omission (RO) and Hit trials during the Uncertain session. White vertical lines denote reward delivery time. ($n = 273$ neurons from 6 mice, each row represents one neuron, and they were sorted by response profiles). Right: mean population responses of outcome monitoring neurons during 1-s window after reward omission or delivery during RO and Hit trials (Wilcoxon signed-rank test, $***P = 1.9209e-17$). **b** Left: trial-averaged go cue-evoked response of value updating neurons in the early (T1 + T2, left) and late (T3, right) periods. White vertical lines denote the stimulus onset. ($n = 88$ neurons from 6 mice, each row represents one neuron, and they were sorted by response profiles). Right: mean population responses of value updating neurons during 1-s window after cue onset in early and late periods (Wilcoxon signed-rank test, $***P = 4.0067e-10$). **c** Venn diagram of the number of neurons identified to perform outcome monitoring (light orange) and value updating (light blue) functions. **d-f** Similar to **a-c**, analysis for the Reversal session. **d** Left: trial-averaged activity of outcome monitoring neurons in response to the air-puff in the early (T1, left) and late (T2 + T3, right) periods. White vertical lines denote air-puff delivery time. ($n = 208$ neurons from 5 mice, each row represents one neuron, and

they were sorted by response profiles). Right: mean population responses of outcome monitoring neurons during 1-s window after air-puff delivery in the early and late periods (Wilcoxon signed-rank test, $***P = 1.9457e-13$). **e** Left: trial-averaged activity of value updating neurons in response to the no-go cue in the early and late periods. White vertical lines denote the stimulus onset. ($n = 348$ neurons from 5 mice, each row represents one neuron, and they were sorted by response profiles). Right: mean population responses of value updating neurons to the no-go cue in the early and late periods (Wilcoxon signed-rank test, $***P = 7.0731e-36$). **f** Venn diagram of the number of neurons identified to perform outcome monitoring (light orange) and value updating (light blue) functions. **g** Venn diagram of the number of outcome monitoring neurons (left) and value updating neurons (right), identified in the Uncertain (light gray) and Reversal sessions (black). Note that 56 neurons recorded in the Uncertain session were not able to be identified in the Reversal session. In box plot: center line, median; box limits, upper and lower quartiles; notch limits, $(1.57 \times \text{interquartile range}) / \sqrt{n}$; whiskers, $1.5 \times \text{interquartile range}$. Outliers are not represented.

Rebuttal Fig. 3 Analyses results of alternative criteria for outcome monitoring and value updating neurons ($\Delta R_{\text{mean}} > 2.5$ s.d.). a-c Identified outcome monitoring neuron

and value updating neuron in Uncertain session. **a** Left: trial-averaged activity of the outcome monitoring neurons in reward omission (RO) and Hit trials during the Uncertain session. White vertical lines denote reward delivery time ($n = 169$ neurons from 6 mice, each row represents one neuron, and they were sorted by response profiles). Right: mean population responses of outcome monitoring neurons during 1-s window after reward omission or delivery during RO and Hit trials (Wilcoxon signed-rank test, $***P = 5.6454e-17$) **b** Left: trial-averaged go cue-evoked response of value updating neurons in the early (T1 +T2, left) and late (T3, right) periods. White vertical lines denote the stimulus onset. ($n = 28$ neurons from 6 mice, each row represents one neuron, and they were sorted by response profiles). Right: mean population responses of value updating neurons during 1-s window after cue onset in early and late periods (Wilcoxon signed-rank test, $***P = 1.1083e-05$). **c** Venn diagram of the number of neurons identified to perform outcome monitoring (light orange) and value updating (light blue) functions. **d-f** Similar to **a-c**, analysis for the Reversal session. **d** Left: trial-averaged activity of outcome monitoring neurons in response to the air-puff in the early (T1, left) and late (T2 + T3, right) periods. White vertical lines denote air-puff delivery time. ($n = 154$ neurons from 5 mice, each row represents one neuron, and they were sorted by response profiles). Right: mean population responses of outcome monitoring neurons during 1-s window after air-puff delivery in the early and late periods (Wilcoxon signed-rank test, $***P = 2.2721e-09$). **e** Left: trial-averaged activity of value updating neurons in response to the no-go cue in the early and late periods. White vertical lines denote the stimulus onset. ($n = 158$ neurons from 5 mice, each row represents one neuron, and they were sorted by response profiles). Right: mean population responses of value updating neurons to the no-go cue in the early and late periods (Wilcoxon signed-rank test, $***P = 9.8058e-26$). **f** Venn diagram of the number of neurons identified to perform outcome monitoring (light orange) and value updating (light blue) functions. **g** Venn diagram of the number of outcome monitoring neurons (left) and value updating neurons (right), identified in the Uncertain (light gray) and Reversal sessions (black). Note that 56 neurons recorded in the Uncertain session were not able to be identified in the Reversal session. In box plot: center line, median; box limits, upper and lower quartiles; notch limits, $(1.57 \times \text{interquartile range}) / \text{sqrt}(n)$; whiskers, $1.5 \times \text{interquartile range}$. Outliers are not represented.

6. Difference of RO to hit across phases- Is the difference present already in the early phase or only in the late phase? Is it caused by a decrease in the hit or an increase in RO? Does exposure to RO decrease the value of Hit? Or would additional exposure to stable tasks lead to a devaluation of HIT outcomes as well?

Thanks for pointing this out. We compared the outcome response of Hit trials and RO trials in the T1, T2 and T3 phases. The difference in response to RO and reward emerged in T2 and increased further in T3 (**Rebuttal Fig. 10d**). Such difference was mainly

caused by the increase in RO response, as both RO and Hit response increased with time in the Uncertain session (**Rebuttal Fig. 10b, c**). In contrast to the Uncertain session, the response to the reward outcome (in the Hit trials) decreased over time during the Stable session (**Rebuttal Fig. 10a**), but the go stimulus evoked response did not change across T1-T3 in the Stable session (**Fig. 5d** in the manuscripts).

Rebuttal Fig. 10 Outcome evoked responses in the Hit and RO trials across T1-T3 in the Uncertain session. **a** Mean response of all neurons during 1-s window after reward delivery during Hit trials in the T1-T3 phases of the Stable session (n = 549 neurons from 6 mice, Friedman test with post hoc Bonferroni multiple comparisons. T1 vs. T2: $***P = 7.3873e-07$; T1 vs. T3: $***P = 4.1647e-05$). **b** Similar with **a**, except for Hit trials of Uncertain session (n = 549 neurons from 6 mice, Friedman test with post hoc Bonferroni multiple comparisons. T1 vs. T3: $**P = 0.0024$; T2 vs. T3: $***P = 7.3873e-07$). **c** Mean responses of all neurons during 1-s window after reward omission during RO trials in the T1-T3 phases of the Uncertain session (n = 549 neurons from 6 mice, Friedman test with post hoc Bonferroni multiple comparisons. T1 vs. T3: $***P = 8.6757e-07$; T2 vs. T3: $***P = 3.8026e-04$). **d** The difference in outcome evoked responses between RO and Hit trials in the T1-T3 phase of Uncertain sessions. The symbol * denotes significant differences between different phases (Friedman test with post hoc Bonferroni multiple comparisons, T1 vs. T3, $*P = 0.0217$). The symbol \$ denotes a significant difference compared with 0 (one-sample Wilcoxon signed-rank test. T2, $$$$P = 7.0513e-04$; T3, $$$$P = 2.2207e-06$). In box plot: center line, median; box limits, upper and lower quartiles; notch limits, $(1.57 \times \text{interquartile range}) / \sqrt{n}$; whiskers, $1.5 \times \text{interquartile range}$. Outliers are not represented

7. “Line 189: Although the dual-function neurons only took a small portion here (78 out of 549 neurons), the decoding accuracy dropped significantly when they were removed from the population (Fig. 3h), indicating that they played an important role in computing outcomes and tracking experience history. The dual-function neuron might be a natural computation and storage element.” - Does this really indicate history tracking? Does a high classification precision not only indicate a degree of immediate decoding (Fig. 3g higher value in the current trial if RO) but does not mean this is part of a “tracking” mechanism? Or as storing the information? Do they change over time, and the difference increases?

Thanks for pointing it out. We should have been more specific and precise in describing

the results. We did find the number of dual-function neurons increases with time in the Uncertain session (**Supplementary Fig. 10** in the revised manuscript). The difference in RO and reward outcome (in the Hit trials) evoked response of dual-function neurons also decreased with time (**Rebuttal Fig. 11**), while the difference of go cue evoked response of dual-function neurons between T3 and T1 was larger than that of that between T2 and T1. The information of previous unexpected outcome was indeed transformed to the value representation and stored in ACC, perhaps in the form of the recruited number of dual-function neurons and their activity.

Supplementary Fig. 10 Progressive recruitment of feedback-guided value iteration neurons in the Uncertain session. **a** Trial-averaged activity of all neurons in response to the go cue in the T1 (left) and T2 (right) phases. White vertical lines denote stimulus onset. Each row represents one neuron and they were sorted by response profiles. **b** Mean population responses to the go cue of recruited value updating neurons (marked by blue vertical bars adjacent to heat map in **a**) in the T1 and T2 phases (Wilcoxon signed-rank test). **c** Venn diagram describing the numbers of outcome monitoring neurons (light orange, identified in Fig. 3g) and value updating neurons recruited in the T2 phase (light blue). **d-f** Similar to **a-c**, but the analysis was carried out for T1 and T3 phases. **g** Venn diagram of the number of dual-function neurons (simultaneously performing both outcome monitoring and value updating functions) identified in the T2 (light gray) and T3 (black) phases. $***P < 10^{-3}$. In box plot: center line, median; box limits, upper and lower quartiles; notch limits, $(1.57 \times \text{interquartile range}) / \sqrt{n}$; whiskers, $1.5 \times \text{interquartile range}$. Outliers are not represented. See Table S1 for detailed statistics.

Rebuttal Fig. 11 Responses of dual-function neurons in the Uncertain session. **a** The difference of outcome evoked responses between RO and Hit trials in the T1-T3 phase of Uncertain sessions. The symbol * denotes significant differences between different phases (n = 78 neurons for 6 mice, Friedman test with post hoc Bonferroni multiple comparisons, T1 vs. T2, $**P = 0.007$). The symbol \$ denotes a significant difference compared with 0 (one-sample Wilcoxon test, T1, $$$$P = 3.8439e-06$; T3, $$P = 0.0136$). **b** The difference of go cue evoked responses between T1 and T2 phase (T1-T2), and between T1 and T3 phase (T1-T3) of Uncertain sessions (n = 78 neurons for 6 mice, Wilcoxon signed-rank test $***P = 7.9063e-11$). In box plot: center line, median; box limits, upper and lower quartiles; notch limits, $(1.57 \times \text{interquartile range}) / \text{sqrt}(n)$; whiskers, $1.5 \times \text{interquartile range}$. Outliers are not represented.

We modified our description in the revised manuscript as (Line 370-376)

“On the other hand, if more error events were detected in the next few trials, it would trigger the recruitment of more and more units to track and convert the error signal, and therefore push the rapid shift of the value representation and decision strategy. During the Uncertain session, with the accumulation of occasional unexpected feedback, the recruitment of dual-function neurons also increased in the T3 phase compared to T2 phase (Supplementary Fig. 10), indicating an accumulative recruitment of iteration units upon errors.”

8. Line 227: *Why are FA responses in the early phase increased, but No-go responses in the late phase decreased? In both, we see that early is higher than late, but this wording indicates different mechanisms. Why cannot both be showing the same (e.g., increased activity immediately after Reversal? FA trials are a subset of No-go cues trials, after all). I am not sure if there is enough evidence to state “the No-go stimulus-evoked response was significantly reduced in the late period due to the cumulative devaluation of the previous reward-associated stimulus.” Can the authors clarify?*

Thanks for pointing out the possible wording confusion. We have changed the description of this result in the revised manuscript (Line 229-235) as suggested by the reviewer:

“The punishment evoked population response (1 s since air-puff delivery) was

stronger in the early period (Fig. 4g, h), revealing that a more prominent error signal was generated when the outcome was most unexpected.

...

On the other hand, the no-go stimulus-evoked response was also stronger in the early period, because the previous reward-associated stimulus was devaluated by repeated punishment (Fig. 4j, k).”

The tone evoked responses in both the CR and FA trials were much higher in the early stage in the Reversal session (**Rebuttal Fig. 7**), therefore we concluded that the no-go stimulus evoked response was significantly reduced in the late period. It was likely due to the receiving air puff punishment in the early phase of the Reversal session, which led to the devaluation of the 3 kHz tone.

Rebuttal Fig. 7 Significant decrease of selectivity of both behavior decision and neuronal response across T1-T3 in the Reversal session. **a** Left: performance discrimination (d') across the T1-T3 phases of Reversal session ($n = 5$ mice, one-way repeated measures ANOVA with post hoc Bonferroni multiple comparisons. $F(1.955, 7.820) = 8.565$, $P = 0.0109$. T1 vs. T3: $*P = 0.0326$. Right: performance of Hit and FA rate in T1-T3 phases. **b** Mean responses to go and no-go cues across the T1-T3 phases of the Reversal session ($n = 493$ neurons from 5 mice, Friedman test with post hoc Bonferroni multiple comparisons. no-go cue: T1 vs. T2: $***P = 2.9867e-16$; T1 vs. T3: $***P = 1.1685e-19$; go vs. no-go: T1, $***P = 8.4422e-07$; T3, $*P = 0.0144$). **c** Stimulus selectivity index (SI) of individual neurons in the T1-T3 phase of Reversal sessions ($n = 493$ cells from 5 mice, one-sample Wilcoxon signed-rank test, compared with 0: T1, $***P = 1.180e-15$; T2, $***P = 3.726e-10$). Color dots indices neurons with significant SI in each phase ($P < 0.05$, permutation test). Error bars indicate the standard error of the mean. In box plot: center line, median; box limits, upper and lower quartiles; notch limits, $(1.57 \times \text{interquartile range}) / \sqrt{n}$; whiskers, $1.5 \times \text{interquartile range}$. Outliers are not represented.

9. Fig 5j,k why is t-1 opposite to t-2?

Thanks a lot for pointing out the typo. We have corrected the title of Fig. 4g as “T1” vs “(T2+T3)”.

Minor Comments:

1. The manuscript is well written. However, I believe it will benefit from thorough

proofreading. At times, the concepts are quite difficult to parse out and grasp and are presented in a rather convoluted way.

We have modified the description and the schematic of our working model (Fig. 1a) and modified the description of the results and discussion. We hope it would be more comprehensive.

2. The configuration of recording ACC neurons was not mentioned clearly – I believe an imaging configuration in Fig. 1g would be helpful.

Thanks for pointing out the confusing point.

(1) We modified **Fig. 1d** in the manuscript to show the configuration of ACC recording, and added explanation of imaging procedures in the Methods in the revised manuscript (Line 483-485) as:

“Imaging region of $\sim 250 \times 250 \mu\text{m}^2$ was located lateral to the superior sagittal sinus at a depth of 250 - 400 μm below the pial, corresponding to the posterior dorsal ACC.”

Fig. 1 d Scheme (left) of imaging L2/3 ACC neurons through a cranial window (the grey horizontal bar) and a coronal section (right) of ACC neurons expressing GCaMP6f in the recorded mouse (post-mortem). The red dotted lines indicate L2/3 and L5 of the cortex, and the black horizontal line represents the plane of imaging. M2, the secondary motor cortex.

(2) We also added titles for panels in **Fig. 1g**.

3. Line 142: Not sure if the authors can state - based on these results that motivation did not decrease. Please comment and edit.

We stated that the motivation was not changed through the Uncertain session based on the stable lick behavior through the T1-T3 stages. We have addressed as the major comment 1 of Reviewer #2. We have compiled a new **Supplementary Fig. 5** and modified the description in the revised manuscript (Line 142-146) as:

“The experience of RO trials led to an adjustment in the decision strategy, but not a decrease in motivation to perform the task, as the lick latency, the number of anticipatory and the consumption lick rate were stable throughout T1-T3 phases in the Uncertain session and as high as in the Stable session

(Supplementary Fig. 5).”

Supplementary Fig. 5 Consistent performance of anticipatory and consumption licking during the Stable and Uncertain sessions. **a.** The anticipatory licking rate of the Hit trials in the Stable session and T1-T3 phases of the Uncertain session. **b** The number of anticipatory licks (left) and lick latency (right) in each phase ($n = 81\sim 191$ trials from 6 mice. Kruskal-Wallis test with post hoc Bonferroni multiple comparisons). **c** The consumption licking rate relative to the onset of reward delivery in the Stable session and T1-T3 phases of the Uncertain session (left), and the number of total consumption licks in each phase (right, $n = 71\sim 191$ trials from 6 mice. Kruskal-Wallis test with post hoc Bonferroni multiple comparisons). Shading shows the standard error of the mean. In box plot: center line, median; box limits, upper and lower quartiles; notch limits, $(1.57 \times \text{interquartile range}) / \sqrt{n}$; whiskers, $1.5 \times \text{interquartile range}$. Outliers are not represented. See Table S1 for detailed statistics.

4. Fig 3e: The $\Delta F/F$ changes around 5% - how are these values (strongly) significantly different?

The change in population averaged $\Delta F/F$ was around 5%, but the $\Delta F/F$ change of individual neurons could be much larger, varying up to 40%. It could be seen from the heat maps of $\Delta F/F$ of individual neurons in Fig.3a and 3d in the manuscript.

5. Line 204: how is 1d relevant there?

Thanks for pointing out the typo. It should be Fig. 1i. We have fixed it.

6. Line 216/Fig 4e: immediate response to the stimulus itself seems similar, but there is a later decrease (around 0.25s post stimulation) – how do the authors explain this?

The response immediately after the onset of tone (0-0.25s) might be more influenced by tone itself, but the response afterward might be encoded in the process of value updating, and therefore the difference of response was more prominent in 0.25s post

stimulation.

Reference

- 1 Groblewski, P. A. *et al.* Characterization of Learning, Motivation, and Visual Perception in Five Transgenic Mouse Lines Expressing GCaMP in Distinct Cell Populations. *Front Behav Neurosci* **14**, 104, doi:10.3389/fnbeh.2020.00104 (2020).
- 2 Sun, W. *et al.* The anterior cingulate cortex directly enhances auditory cortical responses in air-puffing-facilitated flight behavior. *Cell Rep* **38**, 110506, doi:10.1016/j.celrep.2022.110506 (2022).

Reviewer #3 (Remarks to the Author):

Major Comments:

1. Throughout the manuscript, an emphasis is placed on the neural responses to the two conditioned stimuli, and changes in neural activity during this period are interpreted to represent changes in the respective incentive values of the cues. However, the latent values associated with each cue are correlated with motor responses (lick bouts) that occur during the cue interval and could equally account for the neural activity. A comparison of neural vs. behavioral response latencies relative to CS onset (Fig. S3) confirmed that neural responses occur earlier (median, 230 vs. 300 ms). However, this difference is overstated in the text (“...onset of go tone evoked response was much earlier than the licking onset”), and it does not rule out the possibility that the neural responses are at least partly motor-related (eg, preparatory activity is well-documented in neighboring areas of the frontal lobe such as M2/MOs). Moreover, the $\Delta F/F$ time series were temporally filtered at 5 Hz prior to analysis, which could affect the result. One solution would be to account for motor and cognitive factors simultaneously by incorporating them into a common statistical model (e.g. multiple linear regression) with the neural responses as the output variable. Another solution could be to amend the behavioral task to enforce a delay or trace period following the sound cue, during which time licking is penalized.

We appreciate that the reviewer pointed out the critical point that licking behavior might determine the cue-evoked response and we are very grateful for the valuable suggestion to use GLM to estimate the contribution of licking to the neural response. We analyzed the relationship between neuron activity and licking behavior in the Uncertain and Reversal sessions in the first part of the rebuttal (addressed together with the major comment 2b of reviewer #2) and added the analysis results in **Supplementary Fig. 3** in the revised manuscript. We found that the cue evoked response in ACC neurons was not closely correlated with the number or onset of licks during the stimulus window and the contribution of licking to the stimulus evoked response was much smaller than other cognitive factors.

Regarding the concern about using a 5 Hz filter for $\Delta F/F$ trace, we also filtered $\Delta F/F$ trace with median filter of 3 points, and the response onset time was not different from that of the $\Delta F/F$ trace filtered with 5 Hz (**Rebuttal Fig. 12a**). The onset time of $\Delta F/F$ trace with 3-point median filter was also earlier than the onset of the licking (**Rebuttal**

Fig. 12b, 267 ms vs 300 ms).

Rebuttal Fig. 12 The latencies of neuronal response using $\Delta F/F$ trace filtered with median and low-pass filters. **a** The histogram of neural response latency, raw $\Delta F/F$ trace processed with median filtering (yellow, 90ms window) or low-pass filtering (green, 5 Hz) respectively. ($n = 493$ neurons from 5 mice, Wilcoxon rank-sum test. Median, median filtering: 0.267 s; low-pass filter: 0.233 s). **b** The histogram of neural response latency (yellow, median filtering of raw $\Delta F/F$ trace with 90ms filter window) and the first anticipatory lick latency (gray, $n = 493$ neurons from 5 mice, Wilcoxon rank-sum test). All data from the Hit trials in the Stable session

2. The authors claim to have found that

“a subpopulation of individual ACC neurons could reliably integrate the outcome information to the value representation of the stimulus in next-run trials.”

“The error-induced dynamic recruitment of such ACC neurons determined the impact of error signal on the iteration of value updating and the speed of decision adaptation, forming a non-linear feedback driven updating system to secure the appropriate decision switch.”

“Optogenetically suppressing ACC activity did not interfere the behavior performance with the established strategy, but significantly slowed down the feedback-driven decision switching”

These were the main conclusions of the study. However, as far as I can tell, no direct evidence was offered to support any of these claims. For (1) and (2), the claim is about changes between trials—so of course a trial-by-trial analysis would be required. Instead, several analyses were presented where the session is split in thirds, with changes in neural responses by the last third of the session (“T3”) presented as evidence for “next-run” updating. There could be many alternative explanations for changes like these over the course of a behavioral session, e.g. satiety or even drift in the neural representations. For (3), to demonstrate that learning was hampered, it would be necessary to compare learning rates in some way (number of trials-to-criterion, etc.), but no such comparison is given. Furthermore, Fig. 6h focuses on the FA rate to show some modest impairment in task performance—what about hit rate? From the example

session, the most obvious difference between Stable and Reversal sessions is that hit rate approaches zero (regardless of opsin or control group). The measures presented in this figure could appear “cherry-picked” because 6b compares hit-rates, while 6c compares d' and hit-rate, and finally 6d compares only d' and the lick-responses to the CS+ (stable sessions) vs. the new CS- (Reversal). It would be better to pick one metric (d' , FA rate, hit rate, etc.) and use it for comparisons throughout—or better, to present all three for 6b,e,h.

(1) We were very grateful for the suggestion to use the reinforcement learning model to estimate the trial-by-trial change in value during the task. It indeed made the result much more understandable. We use the State-Action-Reward-State-Action (SARSA) model to estimate the trial-by-trial iteration of stimulus-action value (Q) of 3 kHz tone and the prediction error (Δr) in the Stable, Uncertain and Reversal sessions. The results were summarized in a new **Supplementary Fig. 11 and Fig. 4o** in the revised manuscript.

The Q-value was updated after each trial by the previous Q-value, the detected prediction error (Δr), and the learning rate (α). The model was modified so that the learning rate was no longer a constant, but dynamically regulated by the history of the learning rate and the prediction error ($\alpha_t = \theta' \alpha_{t-1} + \theta |\Delta r_{t-1}|$, **Fig. 4o**). The function of the Q-P relationship was derived from the experimental data, and used to estimate the licking probability to simulate the action sequence according to the action-outcome rule of each session and iterated Q-values (**Supplementary Fig. 11a**). The simulated action sequence was repeated for 3,000 times. The Q-value and lick probability of the simulated action sequence through T1-T3 phases in the Uncertain and Reversal sessions did not differ from the result obtained from the experiment (**Supplementary Fig. 11b-c**). The results showed that the learning rate was much higher in the Reversal and Uncertain sessions than that in the Stable session (**Figure 4o**). The estimated trial-by-trial Q-value and lick probability were also very similar between the experiment data and the simulated action sequence in the first 10 trials of the Reversal session, indicating that the modified SARSA well captured the process of feedback-guided behavior adaptation (**Supplementary Fig. 11d**). The model estimated Q-value (Q) and absolute value prediction error ($|\Delta r|$) were well correlated with the stimulus and outcome evoked response of the dual-function neurons. It suggested a non-linear mechanism for how dual-function ACC neurons transformed the prediction error to renewed value

representation.

Fig. 4 o Left: schematic of the SARSA model. Right: average learning rate of the value iteration of the 3 kHz tone in the Stable, Uncertain, and Reversal sessions. (The Reversal session includes only the first 10 trials. $n = 5$ mice, one-way repeated measures ANOVA with post hoc Tukey's multiple comparisons).

Supplementary Fig. 11 SARSA model recapitulates behavioral and neural data. a

Schematic diagram of the process of using a SARSA model to simulate trial-by-trial Q-value and action sequence according to the value iteration and QP transformation function.

b The licking probability and Q-value across T1-T3 phases in the Uncertain session of the experiment group and simulated group. Blue shade: 95% distribution of simulated data (3,000 repeats). Mean experiment results (red line) fell within the 95% distribution of the simulated data. (experiment: $n = 6$ mice, stimulated: $n = 3,000$ repeats.) **c** Similar to **b**, except for the Reversal session. **d** The licking probability of the experiment group and simulated group in the first 10 trials of 3 kHz stimulus in the Reversal session. The lick probability of each animal was calculated by the probability of over 5 adjacent trials. The lick probability of the simulated group was calculated by the choice of over 3000 repeats. The decay of the licking probability was fit with an exponential function for each group. **e**

Left: relationship between the model estimated Q-value and the mean population responses to the stimulus in the first 10 trials of 3 kHz stimulus in the Reversal session. (n = 5 mice, 10 trials/animal). Right: relationship between the model estimated prediction error (Δr) and the mean population responses to the air puff in FA trials in the Reversal session (n = 5 mice, 2~5 trials/animal). The responses of dual-function neurons correlated to the model estimated Q-value and prediction error, with much better correlation and steeper slope of linear correlation than the rest of the population. Data points from the one mouse were marked with the same color. The data points corresponding to the responses of dual-function neurons and the rest of the population were marked with magenta and cyan outlines, respectively. *P* values test the significance of the correlation

We added description of these results in the revised manuscript (Line 241-254) as:

“To estimate how it happened, we built a reinforcement learning model to estimate the trial-by-trial change of incentive value (Q) of 3 kHz tone updated with the feedback of prediction error (Δr). The learning rate (α) in this modified state-action-reward-state-action (SARSA) model was also trial-by-trial modulated by the prediction error and the previous learning rate ($\alpha_t = \theta' \alpha_{t-1} + \theta |\Delta r_{t-1}|$). The model well predicted the change of 3 kHz-associated value and licking probability (Supplementary Fig. 11). The yielded learning rate was near zero in the Stable session, but higher in the Uncertain and early Reversal sessions (Fig. 4o). Notably, the learning rate in the early Reversal session was 2-fold higher than that in the Uncertain session, similar to the 2-fold increase of recruited dual-function neurons in the population. Each dual-function neuron could work as a parallel computation unit for feedback-guided value iteration. Dynamic recruitment of them could regulate the impact of error feedback and the 253 learning rate of value updating and decision switch, in order to achieve flexible, yet robust enough, feedback-guided value and decision update.”

We added description of SARSA model in the revised manuscript (Line 628-645) as:

“A modified State-action-reward-state-action (SARSA) model was applied to estimate the trial-by-trial incentive value of the 3 kHz tone (Q) and prediction errors (Δr), with the choice of action as lick or no-lick (A_{lick} or A_{nolick} , respectively).

$$Q(S,A) \leftarrow Q(S,A) + \alpha \Delta r = Q(S,A) + \alpha (R - Q(S,A))$$

The learning rate (α) was also updated trial-by-trial with a transfer rate of θ' and θ for the learning rate in the previous trial and the prediction error.

$$Q(A_{j+1}) \leftarrow Q(A_j) + \alpha_j \Delta r^j = Q(A_j) + (\theta' \alpha_{j-1} + \theta \alpha \Delta r^{j-1}) * \Delta r^j$$

The reward of the action (R) was set to 1 (Hit), 0.1 (Miss) and -0.1 (RO) in the

Stable and Uncertain session, and -2 (FA) and 0.1 (CR) in the Reversal sessions, respectively. The initial state was set as $\alpha_0 = 0.1$, $Q_0^{lick} = 0.98$ and $Q_0^{nolick} = 0.02$, the transfer rate was set as $\theta' = 0.6$ and $\theta = 0.08$ for all sessions. Simulated task performance was repeated for 3,000 times with the same action-outcome rule in each session. The action sequence was generated based on the trial-by-trial licking probability (P), which was estimated by the value of the two-action choice ($Q_i = (\{Q_n^{lick}\}, \{Q_m^{nolick}\})^T$), according to the QP transformation function with a linear proportional scaling index (β) derived from the experimental performance ($\beta = 2.3$ for the Stable and Uncertain sessions and $\beta = 1.68$ in the Reversal session, Supplementary Fig. 11).

$$P = f(Q_i, \beta) = \frac{e^{\beta Q_i}}{\sum_{j=1}^n e^{\beta Q_j}} \quad "$$

(2) Thanks a lot for suggesting a better way to demonstrate the difference of strategy switch. We performed the analysis and added the results as a new **Supplementary Fig. 12** in the revised manuscript. We calculated change of FA rate in the Reversal session using an average window of 5 trials, and determined the reversal point as the time when the FA rate reached 25%. The reversal point was much delayed when the activity of ACC neurons was suppressed.

Supplementary Fig. 12 Optogenetic inhibition of ACC delayed decision switch in the Reversal session. **a** Learning curve (FA rate) in the Reversal session. The thick grey and orange lines represent the mean performance values of mCherry ($n = 6$ mice) and eNpHR ($n = 4$ mice) groups, and thin lines represent each animal's performance, respectively. The FA rate was calculated by a sliding window of 5 trials. **b** The number of trials required for each mouse to reach the switching threshold (FA < 25%, indicated by the red dashed line in **a**) for the first time (unpaired t-test). $**P < 10^{-2}$. Error bars indicate the standard error of the mean. See Table S1 for detailed statistics

(3) Regarding Fig. 6, the go cue was presented as completely different tones in the Stable and Reversal sessions. The 12 kHz tone was presented as go cue in the Reversal session, but was also presented as no-go cue in the Stable session. The near-zero Hit rate in the Reversal session was similar to the very low CR rate in the Stable session.

Therefore, the lick rate to the 12 kHz tone was very low in both the Stable and Reversal sessions, because the Hit rate in the Reversal session and the CR rate in the Stable session were both very low. The most obvious change between the Stable and Reversal sessions was that the lick probability in response to 3 kHz tone, corresponding to the Hit rate and FA rate in the Stable and Reversal session respectively, decreased rapidly in the Reversal session. Therefore, we compared the change in FA rate in the Reversal session.

Our description may be somewhat confusing in the original manuscript, and we have modified the figure panels, legends and text in the revised manuscript. We compared the same measures of performance, d' and the lick probability in response to 3 kHz, in all sessions to avoid the confusion that licking upon the same tone was respectively considered as Hit in the Stable and Uncertain session, but FA in the Reversal session.

Fig. 6 Optogenetic suppression of ACC impairs decision switching without disrupting the implementation of established decision rules. **a** Top left: scheme of optogenetic suppression of the ACC neurons. Right: representative images of eNpHR expression and the position of the implanted fiber above ACC. **b** Suppressing ACC activity in the Stable session does not harm the task performance. 589 nm laser suppression is delivered through bilateral optic fibers during the 3 kHz stimulus window in randomly

selected 50% of go trials (left). There was no significant change in behavior performance for d' (middle) and 3 kHz licking probability (Hit rate, right) during ACC inhibition (mCherry: $n = 6$ mice; eNpHR: $n = 4$ mice. Two-way repeated measures ANOVA, within-group comparisons: paired t-test with post hoc Bonferroni multiple comparisons; between-group comparisons: unpaired t-test with post hoc Bonferroni multiple comparisons). **c-e** Suppressing ACC activity in the Uncertain session blocked decision switch (mCherry: $n = 6$ mice; eNpHR: $n = 4$ mice. Two-way repeated measures ANOVA, within-group comparisons: paired t-test with post hoc Bonferroni multiple comparisons; between-group comparisons: unpaired t-test with post hoc Bonferroni multiple comparisons). **c** 589 nm laser is delivered during the 3 kHz stimulus window in the Uncertain session but not in the Stable session. **d** Performance of a representative animal in the mCherry (top) and eNpHR mice (bottom) in Stable (left) and Uncertain (right) sessions. **e** Behavior performance for d' (left) and 3 kHz licking probability (Hit rate, right). mCherry: $n = 6$ mice; eNpHR: $n = 4$ mice. Two-way repeated measures ANOVA, within-group comparisons: paired t-test with post hoc Bonferroni multiple comparisons; between-group comparisons: unpaired t-test with post hoc Bonferroni multiple comparisons). **f-h** Suppressing ACC activity in the Reversal session slowed down the decision switch. Similar to **c-e**, except optogenetic suppression was delivered during the 3 kHz stimulus in the Reversal session. $*P < 0.05$, $**P < 10^{-2}$, $***P < 10^{-3}$. Error bars represent the standard error of the mean. See Table S1 for detailed statistics.

3. A conceptual model for value updating is offered in Figure 1a, but no efforts were made to demonstrate a match between this model and the neurobehavioral data. A rigorous behavioral model could help clarify some of the most important issues in this study. For example, a reinforcement learning model (eg, Q-learning) could provide trial-by-trial estimates of the incentive values ascribed to each cue, as well as the associated prediction errors, to be used (in conjunction with motor variables, etc) in the statistical assessment of factors contributing to the neural activity.

Thank you very much for suggesting the Q-learning model to estimate the trial-by-trial updating of value and prediction error during the task performance. We used a modified SARSA model to simulate the process of non-linear feedback-driven iteration of values. It was also addressed as a common response to the major comment 1.

Minor Comments

1. The writing was very difficult to understand at times.

The terms “value” and “valence” are used interchangeably throughout, but they do not mean the same thing. Value is graded and valence reflects positive or negative affective impact (or association with approach/avoidance) irrespective of magnitude.

Thanks for pointing out the incorrect description. We changed description of “valence”

in the text into “value”, which was more accurate for our conclusion. We also modified other confusing descriptions of the manuscript.

2. *Figure 4g: The distribution of consummatory lick-times relative to the reward has a distinctive multi-modal shape (with peaks at ~ 150 and 500 ms and a prominent trough at ~300 ms). Keeping in mind that the typical lick pattern of C57/Bl6 mice is very stereotyped with a frequency of ~7 Hz within each bout and a corresponding inter-lick interval of ~140 ms (for example, see Raymond et al., 2018), how would the authors explain this? Could there be a problem with the method of lick detection?*

(1) We re-examined the filtered trace of the lick rate in **Supplementary Fig. 4g** and we found that the rightward shift resulting from the filtering process was not corrected in the original manuscript and we have corrected it in the revised version (see **Supplementary Fig. 5c** in the revised manuscript).

Supplementary Fig. 5 c The consumption licking rate relative to the onset of reward delivery in the Stable session and T1-T3 phases of the Uncertain session (left), and the number of total consumption licks in each phase (right, $n = 71\sim 191$ trials from 6 mice. Kruskal-Wallis test with post hoc Bonferroni multiple comparisons). Shading shows the standard error of the mean. In box plot: center line, median; box limits, upper and lower quartiles; notch limits, $(1.57 \times \text{interquartile range}) / \sqrt{n}$; whiskers, $1.5 \times \text{interquartile range}$. Outliers are not represented. See Table S1 for detailed statistics.

(2) Licks were detected by a capacitance sensor and recorded at 1 kHz by Arduino controlled by Arcontrol system¹. We have saved videos of all recorded sessions and re-examined them offline. We found that the licks were faithfully recorded by our system (**Rebuttal Fig. 12a**). The drop in consumption lick rate around 150 ms was probably due to the swallowing of the sucrose liquid. A similar pattern of lick rate decrease due to liquid consumption has also been observed in other studies^{2,3}. We quantified the inter-lick intervals of reward consumption (within 0.25s of valve opening) and continuous licking after consumption (0.25s after valve opening). The distribution of inter-lick interval of those two stages was different (**Rebuttal Fig. 12b**), and the inter-

lick interval of continuous licking after consumption was 143 ± 97.55 ms (median \pm s.d.), close to stereotyped 7-Hz licking bouts.

Rebuttal Fig. 12 Consumption licking during task. **a** Side view of a behavioral mouse recorded with a video camera, showing all the frames in the middle of the first and second lick of the first row of the raster plot below. The first frame (f0) is aligned to trigger the first lick of the reward. The red triangle indicates the tongue appears. **b** Histogram of the distribution of lick intervals after the reward delivery for the period within 0.25s (cyan, median: 250 ms) and post 0.25s (magenta, median: 143 ms). The different colored triangles at the top represent the median lick intervals of each period. Data from the Hit trials in the Uncertain session

Reference

- 1 Chen, X. & Li, H. ArControl: An Arduino-Based Comprehensive Behavioral Platform with Real-Time Performance. *Front Behav Neurosci* **11**, 244, doi:10.3389/fnbeh.2017.00244 (2017).
- 2 Tsutsumi, S. *et al.* Modular organization of cerebellar climbing fiber inputs during goal-directed behavior. *Elife* **8**, doi:10.7554/eLife.47021 (2019).
- 3 Banerjee, A. *et al.* Value-guided remapping of sensory cortex by lateral orbitofrontal cortex. *Nature* **585**, 245-250, doi:10.1038/s41586-020-2704-z (2020).

REVIEWER COMMENTS

Reviewer #1 (Remarks to the Author):

The revised manuscript addressed my concerns to the original manuscript. However, I think Rebuttal Fig1 should be included as a supplemental figure in the manuscript. I also did not find Sup Fig 13 was cited nor discussed in the main manuscript. I suggest this figure to be integrated to the main Fig 6.

Reviewer #2 (Remarks to the Author):

The authors satisfactorily addressed most of my concerns. They summarised, with additional data and explanation, points that I raised earlier. I appreciate the authors have done a thorough job; however, I do have minor points that I must ask for clarification about. They relate to my earlier comments (listed below again).

Previous Major Comment 1:

I think the issue of the low amount of RO and UR trials is not fully addressed. The authors seem to focus on the behavioural response to these trials when they argue that T1-T3 responses were consistent. When looking at neural responses, it is shown in Fig. 2 that there are significant differences between T1 and T3, which is an important finding of the manuscript. The fact that differences between phases are with only around three trials per phase (~10 trials total for all 3 phases) is not properly addressed. This requires acknowledgement and some discussion.

Previous Major Comment 3:

As I understood, ACC neuron responses are downregulated in uncertain and reversal sessions (again mentioned in answer to comment 4), but the authors write, "Similar upregulated response and selectivity was also observed in ACC (Supplementary Fig. 2b and 9f), indicating that the ACC and other frontal cortical areas including OFC, forming a larger network providing necessary information to the decision-making process." Either there are different subpopulations in ACC doing opposite things, or there seems to be a discrepancy between statements, which is confusing. Please clarify why this might be the case and add a discussion.

Previous Major Comment 4:

I like the approach of identifying ACC-A1 projections by labelling as well as imaging A1 activity to address this question. Unfortunately, however, there was no combination of both (there is only structural information about which cells project from ACC to A1, but no functional information about the activity of these specific cells, and there is only general A1 activity information, but not if A1 cells that receive direct input behave differently from the others). Together with the fact that only a really low number of projecting neurons could be identified, I am not sure to what extent speculations about direct influence from ACC to A1 can be drawn. Please discuss this and tone down the conclusions.

Previous Minor comment 4:

Not sure how individual neuronal variability contributes to the significance of two similar mean values since individual variance would make a clear separation of distributions of both groups less likely and, therefore, reduce the likelihood of significance?

Previous Minor comment 6:

The authors addressed the concern with a speculated explanation, which might be correct but is not proven. Also, I am not sure if it is logical to assign this difference to potential value updating since this would rather happen after the outcome experience than immediately after the stimulation (no outcome happened for this stimulus, yet which would be driving a value update).

We thank the reviewers for their very constructive feedback and great suggestions. We have revised the manuscript as suggested. All the comments are addressed as below:

REVIEWER COMMENTS

Reviewer #1 (Remarks to the Author):

The revised manuscript addressed my concerns to the original manuscript. However, I think Rebuttal Fig1 should be included as a supplemental figure in the manuscript. I also did not find Sup Fig 13 was cited nor discussed in the main manuscript. I suggest this figure to be integrated to the main Fig 6.

We have integrated **Supplementary Fig. 13** to main **Fig. 6** and added a brief description of the results in the main text. We have also move rebuttal Fig 1 to supplementary Figure 13.

We add the description in the revised manuscript (Line 306-308) as:

“Similar effect was also observed when ACC activity was ontogenetically suppressed during the response window when unexpected outcomes were experience in the RO, UR and FA trials (Fig. 6i-o).”

Reviewer #2 (Remarks to the Author):

The authors satisfactorily addressed most of my concerns. They summarised, with additional data and explanation, points that I raised earlier. I appreciate the authors have done a thorough job; however, I do have minor points that I must ask for clarification about. They relate to my earlier comments (listed below again).

Previous Major Comment 1:

I think the issue of the low amount of RO and UR trials is not fully addressed. The authors seem to focus on the behavioural response to these trials when they argue that T1-T3 responses were consistent. When looking at neural responses, it is shown in Fig. 2 that there are significant differences between T1 and T3, which is an important finding of the manuscript. The fact that differences between phases are with only around three trials per phase (~10 trials total for all 3 phases) is not properly addressed. This requires acknowledgement and some discussion.

We agree that the number of RO and UR trials was low, and it was the purpose of the Uncertain session to test the impact of low probability of unexpected outcomes on the behavioral and neural response. Although the changed was induced by a very small number of RO/UR trials, the difference of go cue evoked response in T1/T2/T3 phases were calculated from ≥ 12 trials per phase in each animal. The difference of outcome

evoked response was calculated by comparing all RO trials (≥ 6 trials/animal) and Hit trials (≥ 23 trials/animal).

In **Supplementary Fig. 7**, we compared the RO and Hit trials in each phase separately. We agree with the reviewer that it was too bold to draw a conclusion based on only ~ 3 RO trials per phase here. We thank the reviewer to raise this issue and have noted the small number of RO trials in each phase in the description of **Supplementary Fig. 7**. We also removed **Supplementary Fig. 10** and tune down the discussion about the result in the main text.

Previous Major Comment 3:

As I understood, ACC neuron responses are downregulated in uncertain and reversal sessions (again mentioned in answer to comment 4), but the authors write, “Similar upregulated response and selectivity was also observed in ACC (Supplementary Fig. 2b and 9f), indicating that the ACC and other frontal cortical areas including OFC, forming a larger network providing necessary information to the decision-making process.” Either there are different subpopulations in ACC doing opposite things, or there seems to be a discrepancy between statements, which is confusing. Please clarify why this might be the case and add a discussion.

I am sorry that the description here was confusing. We should have specified that change of response. We have revised the discussion to be more accurate as following (Line 327-330):

“Similar upregulated response and selectivity after reverse learning (Re-stable session) was also observed in ACC (Supplementary Fig. 2b and 9f), indicating that the ACC and other frontal cortical areas including OFC, forming a larger network providing necessary information to the decision-making process.”

Previous Major Comment 4:

I like the approach of identifying ACC-A1 projections by labelling as well as imaging A1 activity to address this question. Unfortunately, however, there was no combination of both (there is only structural information about which cells project from ACC to A1, but no functional information about the activity of these specific cells, and there is only general A1 activity information, but not if A1 cells that receive direct input behave differently from the others). Together with the fact that only a really low number of projecting neurons could be identified, I am not sure to what extent speculations about direct influence from ACC to A1 can be drawn. Please discuss this and tone down the conclusions.

Yes, we have very few retrogradely labelled neurons in ACC and it is hard to conclude the impact of ACC activity on the top-down regulation in A1. We tuned down the

discussion about function of ACC output to sensory areas.

Previous Minor comment 4:

Not sure how individual neuronal variability contributes to the significance of two similar mean values since individual variance would make a clear separation of distributions of both groups less likely and, therefore, reduce the likelihood of significance?

The variability of amplitude of $\Delta F/F$ evoked by go cue was large between individual neurons and the difference of mean values was tested with Wilcoxon signed rank test, which is a paired difference test. The significance of difference tested here was determined by relatively change of the $\Delta F/F$ of each neuron more than the difference of the mean values of the whole population.

Previous Minor comment 6:

The authors addressed the concern with a speculated explanation, which might be correct but is not proven. Also, I am not sure if it is logical to assign this difference to potential value updating since this would rather happen after the outcome experience than immediately after the stimulation (no outcome happened for this stimulus, yet which would be driving a value update).

We speculated that the immediately tone evoked response was due to the sound stimulus itself and the delayed decrease of tone evoked response was due to the value representation, because the renewal of value happened immediately after receiving the outcome in the previous trial but the renewed value representation could only be read out when the cue was presented again. We do not have proof for such speculation and we need better experiment to address this issue in the future.

It should be pointed out that when we prepared the source data and code for revision, we wanted to make sure that we upload the correct source data and analysis code, and we asked another colleague to independently recalculate our data. During this process we discovered that we have made some errors in our analysis code. The mistakes are: (1) We filtered the F trace with a 5Hz low pass filter, but we missed to correct for the phase shift caused by the filtering, and all results were calculated from the shifted traces. (2) We had a piece of junk code in the script for identifying the outcome-monitoring and value-updating neurons (Fig.3 and Fig.4), and we took the mean $\Delta F/F$ of the first neuron as the mean of every neuron. (3) We made a mistake in sorting the history of previous trials (Fig. 5j), and the coefficient of the previous trial calculated by GLM was

wrong, but the difference between the stable and restable sessions was similarly significant. **Although those mistakes did not change the conclusion nor the most of statistical significances of our data, it indeed changed exact numbers of most results panels.**

We immediately started a careful review of all experiment and analysis procedures and recordings of the project when we noticed the mistake. We are sure that no one had any intention of manipulating results or committing any other misconduct. However, this is a very serious oversight on the part of our lab and we are very sorry about it. To make sure that we did the analysis correctly this time, we had three other scientists independently recalculate the results from the raw F traces and the recording of the behavior control device using their own scripts written in different programming languages. We have cross checked the independently calculated results and the analysis code of each other, and also corrected other small typo mistakes in the previous manuscript before we finalize current revised manuscript. The major changes of the analysis are followings:

1. Correct the phase shift introduced by filtering the $\Delta F/F$ trace. This led to the change of the exact number of results related to $\Delta F/F$ and Z-score of neural activity, but the trend and significance of statistics was similar to previous ones.
2. Fix the length of Re-stable session of all animals to 23000 frames (70-88 trials/animal), which was the length of the shortest recording of this session. It led to slight change of static results in **supplementary Figure 9b-e**, including that the no-go cue evoked response was higher in T3 (SFig. 9b), no-go cue evoked response was slightly higher in the late stage (SFig. 9e), the FA rate was slightly higher in post non-FA trials (SFig. 9c, right), and the air puff evoked response was slightly higher in the early stage (SFig. 9d). The changes of those results were partially due to the very low number of FA trials in the Re-stable session.
3. Fix the error in identifying outcome monitoring and value updating neurons, and using $R2 > 2 * R1$ to quantify them. The number of each type of neurons was slightly changed, but the trend and significance of statistics of related results were similar (Figure 3 and 4, supplementary Figure 6, 7 and 10), except that (1) the response of reward in the UR trials in the whole population became slight lower than the response to the reward in the Hit trails in **Supplementary Figure 6b**. However, the response of outcome monitoring neurons was still higher in the UR trials; (2) the response to RO and Hit in the whole population was similar in T1 and T3 phases, and the response to RO in the whole population was stable through T-T3 phases in **Supplementary Figure 7a**. As pointed out by Reviewer 2, the number of RO trial in each phase was very low and we also noted it out in the description of the results.

We also used two other criteria to quantify outcome monitoring and value updating neurons and they exhibited the similar results (please see the other rebuttal file regarding the questions of the previous round of revision).

4. Correct the error in sorting the sequence of trial history in **Figure 5j**. The impact of previous trials (coefficients) all increased in both the Stable and Re-stable session, but the difference between the Stable and Re-stable session became more significant.
5. Correct the quantification licking latency in **supplementary Figure 3e-n**. In the previous manuscript the lick time was binned to the frame time, and therefore the onset time of licks within 0.033 s was binned as the same lick latency. We realized that binning the lick time was not accurate, even though the neural activity was recorded at 30 fps. We recalculated the lick latency with the real lick time, with a sampling rate of 1 kHz. The trials were then sorted, resulting in a change in the number of early and late licking trials, but the significance of statics remained similar.
6. Correct the typo error in plotting Hit rate and FA rate in Figure 2a, 4b and 5a.

This is a very unintentional neglect on the part of our lab and we are very sorry about it. We want to be as open as possible about this situation with the editorial board and our reviewers. **We have marked all the changed part mentioned above in the manuscript in red. We also included another rebuttal file regarding all questions asked by reviewers in the previous round of revision. The results and figure were all recalculated with the corrections mentioned above, and all the changes were also marked in red.**

REVIEWERS' COMMENTS

Reviewer #1 (Remarks to the Author):

The revised manuscript addressed the remaining concerns I had in the previous round of review.

Reviewer #2 (Remarks to the Author):

The two points below are still unclear to me, they must be acknowledged clearly in the main text/discussion.

Major comment 3:

I still do not understand how these statements are not contradictory! In the first one, it says ACC neuron activity is downregulated, and in the second one that it is upregulated. They only added the specification that upregulation happened after reversal, but that is exactly the time when it is said to be downregulated as well.

Minor comment 4:

The issue raised was not if the values are significantly different (which is explained in the answer) but rather about the phrasing in the text that higher variance contributes to said significance.

Reviewer #4 (Remarks to the Author):

As I am joining as an additional reviewer after two rounds of peer review I will keep this brief and focus only on the main strengths and weaknesses of the manuscript as I see them and whether the previous reviewers concerns have been addressed.

- This manuscript represents a substantial body of work combining recording experiments and multiple optogenetic manipulations, and there are some useful and interesting findings in it.

- As previous reviewers have pointed out, there are some limitations of the experiment design, notably that learning is assessed by comparing activity and behavior early vs late in the session, which introduces potential confounds of changes in satiety or motivation. However the authors have done control analyses which suggest that behavioral measures expected to correlate with satiation or motivation do not change over the session which I think is a reasonable way of addressing this issue. The authors have also done a sensible set of analyses in supplementary figure 3 which argue against motor actions (as opposed to cognitive variables) accounting for activity changes in ACC.

- The authors note in the second rebuttal that they detected and fixed a bug in their analysis code which caused various figures to change somewhat. They appear to have handled this in an open and straightforward way and to be making a good faith effort to accurately characterize and portray their data, so I think should not be penalized for this.

- There is some question about how novel the findings are given that there is a substantial literature showing that ACC neurons track outcomes, and the value of different actions or stimuli, as well as previous work showing that ACC tracks the volatility of the environment to regulate learning rates (<https://doi.org/10.1038/nn1954>). However the combination of the physiology recordings with the various optogenetic manipulations in the same task (which appear to show pretty robust effects using sensible stats) does strengthen the overall package.

- The weakest element of the story in my opinion is the claim that there is an active regulation of learning rates in ACC mediated by tracking the recent history of unsigned prediction errors, such

that learning rates are upregulated after surprising outcomes. The claim made in figure 4o is that the learning rates are highest in the reversal session where there are large prediction errors, intermediate in the uncertain session, and lowest in the stable session where prediction errors are small. This panel is based on a simple RL model, however my understanding is that the finding is an inevitable consequence of the design of the model, not of fitting a model that could or could not exhibit this behavior to subject's data. Specifically, the model assumes that unsigned prediction errors increase learning rates, and as there are larger unsigned prediction errors in the reversal session than other sessions this leads to higher learning rates. Critically, none of the parameters that control this aspect of the model's behavior are determined by fitting to subject's data. To make a convincing argument using an RL model that learning rates were controlled by recent unsigned prediction errors, one would have to either compare the goodness of fit of models that do and do not incorporate this feature, or make the strength of this effect a model parameter that determined by fitting the model to subject's data. More broadly, the experiment design precludes making strong claims about different learning rates in the different session types, because in the stable session there are essentially no prediction errors due to the deterministic stimulus-outcome mappings, and in the uncertain and reversal sessions the outcomes are different from each other such that one cannot quantitatively compare learning rates – air puffs might drive faster learning than reward omissions simply because they are more aversive not because the learning rate is different.

Reviewer #2 (Remarks to the Author)

The two points below are still unclear to me, they must be acknowledged clearly in the main text/discussion.

Major comment 3:

I still do not understand how these statements are not contradictory! In the first one, it says ACC neuron activity is downregulated, and in the second one that it is upregulated. They only added the specification that upregulation happened after reversal, but that is exactly the time when it is said to be downregulated as well.

We thank the reviewer for pointing out that the wording should be improved here. Actually, in the manuscript, we referred “after reverse learning” as the Re-stable session rather than the Reversal session. The response in the Re-stable session was indeed very much different from the Reversal session.

The details of what we actually mean is that in the Reversal sessions, the responses to 3 kHz tone were downregulated. After several days of reverse learning, the animal learned to lick upon the new go cue (12 kHz), and the response to new go cue (12 kHz) was increased again in the Re-stable session. The response to the new go cue (12 kHz) in the ACC was higher in the Re-stable session than that to the go cue (3 kHz) in the Stable session. Such change of response to go cues between the Re-stable and Stable session was similar to that in the OFC. We will make it more clearly in the revised manuscript. The changes are as follows:

(Line 326-331):

*“Similar upregulated response and selectivity after **completing** reverse learning (**Re-stable session**) was also observed in ACC (Supplementary Fig. 2b and 9f), indicating that the ACC and other frontal cortical areas including OFC, forming a larger network providing necessary information to the decision-making process.”*

Minor comment 4:

The issue raised was not if the values are significantly different (which is explained in

the answer) but rather about the phrasing in the text that higher variance contributes to said significance.

We are sorry there is a misunderstanding here because we didn't explain it clearly enough before. Actually, this misunderstanding comes from previous rebuttal letters. To better explain this, the complete details about the whole issue is as follows:

The population variation of cue evoked response did not cause the significance of mean difference in **Fig. 3e**, but it was the reason why the amplitude of population averaged response was very small (~5%) while the response of many individual neurons were strong. The related part in the manuscript is:

Fig. 3 Individual dual-function ACC neurons temporally integrate outcome information to the next-run value representation. d Trial-averaged go cue-evoked response of all neurons in the early (T1 +T2, left) and late (T3, right) periods. White vertical lines denote the stimulus onset. Each row represents one neuron, and they were sorted by response profiles. **e** Mean population responses of all neurons during 1-s window after cue onset in early and late periods (Wilcoxon signed-rank test). Inset: Trial-averaged Ca^{2+} response to go stimulus in the early and late periods. Dashed vertical lines denote the stimulus onset. **f** Mean population responses of

identified value updating neurons (marked by light blue bars adjacent to heat maps in d).

A large portion of the neurons exhibited strong response during the cue stimulus, and their response exhibit large variation across the population. The activity of some of them were activated by the cue stimulus, while the others were suppressed (**Fig. 3d**). The responses of the activated and suppressed neurons counterbalanced each other when they were pooled together to calculate the population averaged response. Therefore, the absolute amplitude of mean population response was as small as 5%, whereas many individual neurons exhibited strong responses as large as 40%. We showed the heatmap to demonstrate the response of individual cells (**Fig. 3d**) and the reader can see the diversity of response which was underneath the population mean response.

Neither the variation of population activity nor the absolute value of mean activity caused the significance of response difference between the early and late phases. The significance of difference was tested with a paired comparison for all recorded neurons (**Fig. 3e**). The significance of pair-wise tested here was determined by relatively change of the $\Delta F/F$ of each neuron more than the absolute mean value of the whole population or the variation of values within each group. The variation of cue evoked responses across individual neurons did not affect the significance of response difference between the early and late phases.

It should be pointed out (again) that we only discussed the population variation in the rebuttal letter when the reviewer raised concerns about small absolute mean values. We did not elaborate the description of relationship between the diversity of population response and significance of activity change in the manuscript, since the population diversity of response was clearly demonstrated in the heatmap (**Fig. 3d**) and the statistic test of paired comparison (Wilcoxon signed-rank test) was noted clearly in the figure legend and the statistical report (supplementary Table 1).

As the discussion comes from the previous rebuttal letter but not the manuscript, if the editors and reviewers feel it necessary, we can add the explanations to the manuscript.

Reviewer #4 (Remarks to the Author):

The weakest element of the story in my opinion is the claim that there is an active regulation of learning rates in ACC mediated by tracking the recent history of unsigned prediction errors, such that learning rates are upregulated after surprising outcomes. The claim made in figure 4o is that the learning rates are highest in the reversal session where there are large prediction errors, intermediate in the uncertain session, and lowest in the stable session where prediction errors are small. This panel is based on a simple RL model, however my understanding is that the finding is an inevitable consequence of the design of the model, not of fitting a model that could or could not exhibit this behavior to subject's data. Specifically, the model assumes that unsigned prediction errors increase learning rates, and as there are larger unsigned prediction errors in the reversal session than other sessions this leads to higher learning rates. Critically, none of the parameters that control this aspect of the model's behavior are determined by fitting to subject's data. To make a convincing argument using an RL model that learning rates were controlled by recent unsigned prediction errors, one would have to either compare the goodness of fit of models that do and do not incorporate this feature, or make the strength of this effect a model parameter that determined by fitting the model to subject's data. More broadly, the experiment design precludes making strong claims about different learning rates in the different session types, because in the stable session there are essentially no prediction errors due to the deterministic stimulus-outcome mappings, and in the uncertain and reversal sessions the outcomes are different from each other such that one cannot quantitatively compare learning rates – air puffs might drive faster learning than reward omissions simply because they are more aversive not because the learning rate is different.

We thank the reviewer for pointing out the limitation of the model. We have tried to simulate using conventional RL model with fixed learning rate. It required a learning rate with larger value to fit the Reversal session and a learning rate with a smaller value to fit the Uncertain session, similar to the result of the model with dynamic learning rate. We modified the description of this result to avoid overstate the result of SARSA model.

(Line 240-247):

“The yielded learning rate was higher in the early Reversal session than that in

the Uncertain session (Fig. 4o), similar to the 2-fold increase of dual-function neurons recruited in the Reversal session. One possible function of the dual-function neurons could be that each of them worked as a parallel computation unit for feedback-guided value iteration. Dynamic recruitment of them could regulate the converting of error feedback to value updating and the speed of decision switch, in order to achieve flexible, yet robust enough, feedback-guided value and decision update.”